# Detecting Small Query Graphs in a Large Graph via Neural Subgraph Search

## Abstract

Recent advances have shown the success of using reinforcement learning and search to solve NP-hard graph-related tasks, such as Traveling Salesman Optimization, Graph Edit Distance computation, etc. However, it remains unclear how one can efficiently and accurately detect the occurrences of a small query graph in a large target graph, which is a core operation in graph database search, biomedical analysis, social group finding, etc. This task is called Subgraph Matching which essentially performs subgraph isomorphism check between a query graph and a large target graph. One promising approach to this classical problem is the "learning-to-search" paradigm, where a reinforcement learning (RL) agent is designed with a learned policy to guide a search algorithm to quickly find the solution without any solved instances for supervision. However, for the specific task of Subgraph Matching, though the query graph is usually small given by the user as input, the target graph is often orders-of-magnitude larger. It poses challenges to the neural network design and can lead to solution and reward sparsity. In this paper, we propose NSUBS with two innovations to tackle the challenges: (1) A novel encoder-decoder neural network architecture to dynamically compute the matching information between the query and the target graphs at each search state; (2) A novel look-ahead loss function for training the policy network. Experiments on six large real-world target graphs show that NSUBS can significantly improve the subgraph matching performance.

## 1 Introduction

With the growing amount of graph data that naturally arises in many domains, solving graph-related tasks via machine learning has gained increasing attention. Many NP hard tasks, e.g. Traveling Salesman Optimization (Xing & Tu, 2020), Graph Edit Distance computation (Wang et al., 2021), Maximum Common Subgraph detection (Bai et al., 2021), have recently been tackled via learning-based methods. These works on the one hand rely on search to enumerate the large solution space, and on the other hand use reinforcement learning (RL) to learn a good search policy from training data, thus obviating the need for hand-crafted heuristics adopted by traditional solvers. Such learning-to-search paradigm (Bai et al., 2021) also allows the training the RL agent without any solved instances for supervision. However, how to design a neural network architecture under the RL-guided search framework remains unclear for the task of Subgraph Matching, which requires the detection of all occurrences of a small query graph in an orders-of-magnitude larger target graph. Subgraph Matching has wide applications in graph database search (Lee et al., 2012), knowledge graph query (Kim et al., 2015), biomedical analysis (Zhang et al., 2009), social group finding (Ma et al., 2018), quantum circuit design (Jiang et al., 2021), etc. As a concrete example, Subgraph Matching is used for protein complex search in a protein-protein interaction network to test whether the interactions within a protein complex in a species are also present in other species (Bonnici et al., 2013).

Due to its NP-hard nature, the state-of-the-art Subgraph Matching algorithms rely on backtracking search with various techniques proposed to reduce the large search space (Sun & Luo, 2020; Kim et al., 2021; Wang et al., 2022). However, these techniques are mostly driven by heuristics, and as a result, we observe that such solvers often fail to find any solution on large target graphs under a reasonable time limit, although they tend to work well on small graph pairs. We denote this phenomenon as *solution sparsity*. Such solution sparsity requires the designed model to not only have enough capacity but also to run efficiently under limited computational budget. Another consequence

of solution sparsity is that, there can be little-to-no reward signals for the RL agent under an RL training framework (Silver et al., 2017), which we denote as *reward sparsity*.

In this paper, we propose NSUBS with two means to address the aforementioned challenges. First, we propose a novel graph encoder-decoder neural network to dynamically match the query graph with the target graph and perform aggregation operation only on the query graph to reduce information loss. The novel encoder decouples the intra-graph message passing module (the "propagation" module) that yields state-independent node embeddings, and the inter-graph message passing module (the "matching" module) that refines the node embeddings via subgraph-to-graph matching. Thus, the intra-graph embeddings can be computed only once at the beginning of search for efficient inference. We further advance the inter-graph message passing by propagating only between nodes that either are already matched or can be matched in future by running a local candidate search space computation algorithm at each search state. Such algorithm leverages the key requirement of Subgraph Matching that every node and edge in the query graph must be matched to the target graph, and therefore reduces the amount of candidates from all the nodes in the target graph to a much smaller amount. Compared with a Graph Matching Network (Li et al., 2019) which computes all the pairwise node-to-node message passing between two input graphs, our matching module is able to focus on only the node pairs that can contribute to the solution, and thus is both more effective and more efficient. In addition, we propose the use of sampling of subgraphs to obtain ground-truth subgraph-to-graph node-node mappings to alleviate the reward sparsity issue during training. We design a novel look-ahead loss function where the positive node-node pairs are augmented with positive node-node pairs in future states to boost the amount of training signals at each search state.

Experiments on synthetic and real graph datasets demonstrate that NSUBS outperforms baseline solvers in terms of effectiveness by a large margin. Our contributions can be summarized as follows:

- We address the challenging yet important task of Subgraph Matching with a vast amount of practical applications and propose NSUBS as the solution.
- One key novelty is a proposed encoder layer consisting of a propagation module and a matching module that dynamically passes the information between the input graphs.
- We conduct extensive experiments on real-world graphs to demonstrate the effectiveness of the proposed approach compared against a series of strong baselines in Subgraph Matching.

## 2 PRELIMINARIES

### 2.1 PROBLEM DEFINITION

We denote a query graph as $q = (V_q, E_q)$ and a target graph as $G = (V_G, E_G)$ where $V$ and $E$ denote the node and edge sets. $q$ and $G$ are associated with a node labeling function $L_g$ which maps every node into a label $l$ in a label set $\Sigma$. **Subgraph**: For a subset of nodes $S$ of $V_q$, $q[S]$ denotes the subgraph of $q$ with an node set $S$ and a edge set consisting of all the edges in $E_q$ that have both endpoints in $S$. In this paper, we adopt the definition of non-induced subgraph. **Subgraph isomorphism**: $q$ is subgraph isomorphic to $G$ if there exists an injective node-to-node mapping $M : V_q \to V_G$ such that (1) $\forall u \in V_q, L_g(u) = L_g(M(u))$; and (2) $\forall e_{(u,u')} \in E_q, e_{(M(u),M(u'))} \in E_G$. **Subgraph Matching**: The task of Subgraph Matching aims to find the subgraphs in $G$ that are isomorphic to $q$. We call $M$ a solution, or a match of $q$ to $G$. We call a pair $(q, G)$ is solved if the algorithm can find any match under a given time limit, which we find a challenge for existing solvers on input graphs in experiments especially on large graphs. For solved pairs, the number of found subgraphs by an algorithm is reported.

### 2.2 RELATED WORK

**Non-learning methods on Subgraph Matching** Existing methods on Subgraph Matching can be broadly categorized into backtracking search algorithms (Shang et al., 2008; He & Singh, 2008; Han et al., 2013; 2019; Kim et al., 2021; Wang et al., 2022) and multi-way join approaches (Lai et al., 2015; 2016; 2019; Kankanamge et al., 2017). The former category of approaches employ a branch and bound approach to grow the solution from an empty subgraph by gradually seeking one matching node pair at a time following a strategic order until the entire search space is explored. The multi-way join approaches rely on decomposing the query graph into nodes and edges and performing join

operations repeatedly to combine the partially matched subgraphs to $q$. However, they tend to work well on small query graphs generally with less than 10 nodes (Sun & Luo, 2020), and thus we follow and compare against methods in the former category, whose details will be shown in Section 2.3.

**Learning-based methods for subgraph-related problems**    The idea of designing graph neural networks for graph-graph similarity has been explored, but not Subgraph Matching. GMN (Ling et al., 2020) captures general notions of graph similarity through inter-graph message passing, but outputs a similarity score instead of the discrete matching between 2 graphs. DMPNN (Liu & Song, 2022) uses node to edge conversions to obtain node and edge representations that better preserve isomorphism properties, but outputs approximate subgraph isomorphism counts instead of a discrete matching.

Representation learning methods for Subgraph Matching use a geometric loss to provide soft node-node correspondance scores, stops short of providing a discrete mapping of where the query occurs in the target graph. For example, NMATCH (Lou et al., 2020) learns node embeddings to predict a score for an input subgraph-graph pair indicating whether the subgraph is contained in another graph. ISONET (Roy et al., 2022) extends this idea using edge-edge correspondence scores to rank which query graphs are most likely to appear in the target.

Another direction of research aims to perform subgraph counting (Liu et al., 2020; Chen et al., 2020) supervised on the number of specific substructures, and again lacks an explicitly search strategy and thus falls short of yielding solutions for Subgraph Matching. Researchers tackling consistent subgraph matching (Yuan et al., 2022) handle complex node or edge constraints using a Subgraph Matching subroutine. Hence, advancements to Subgraph Matching can directly benefit such works.

**Efforts on using RL for graph NP-hard problems**    The idea of using RL to replace heuristics in search algorithms for NP-hard graph-related tasks is not new, and we identify three works similar to the present work. (1) GLSEARCH (Bai et al., 2021) detects the maximum common subgraph (MCS) in an input graph pair, which is different from Subgraph Matching which requires the entire $q$ to be matched with $G$, allowing further improvement in the neural network and search design. (2) RL-QVO (Wang et al., 2022) tackles Subgraph Matching via ordering the nodes in the *query* graph as a global pre-processing step before search, which is an orthogonal direction to our approach to select nodes in the target graph computed at each search step.

## 2.3    SEARCH-BASED METHODS FOR SUBGRAPH MATCHING

Due to the NP-hard nature of Subgraph Matching, backtracking search is a naturally suitable algorithm since it exhaustively explores the solution space by starting with an empty match and adding one new node pair to the current match at each step. When the current match cannot be further extended, the search backtracks to its previous search state, and explores other node pairs to extend the match. However, naively enumerating all the possible states in the entire search space is intractable in practice, and therefore existing efforts mainly aim to reduce the total number of search steps

---

**Algorithm 1** Search-based Subgraph Matching.

1: **Input:** Query graph $q$, data graph $G$.
2: **Output:** Matches from $q$ to $G$.
3: Filter: $C \leftarrow$ generate candidate node sets.
4: Order: $\phi \leftarrow$ generate an ordering for $V_q$.
5: Search: `Backtracking`$(q, G, C, \phi, \{\})$.

---

for the backtracking search via mainly three ways (Sun & Luo, 2020): (1) Filter nodes in $G$ to obtain a small set of candidate nodes for each node in $q$ as a pre-processing step before the backtracking search; (2) Order the nodes in $q$ before the search; (3) Generate a local candidate set of nodes in each step of the search based on the current search state. Algorithm 1 summarizes the overall backtracking search based framework for Subgraph Matching. It is worth noting that the first three means correspond to the three steps in the algorithm, and therefore any improvement in any of the three steps can be regarded as orthogonal to each other.

The basic idea of backtracking search is outlined in Algorithm 2. The recursive algorithm starts with an empty node-node mapping, and tries to add one new node pair to the mapping $M$ at each recursive call. The action is the new node pair $(u_t, v_t)$, where $u_t \in V_q$ is selected according to the

heuristic-based ordering $\phi$, and $v_t \in V_G$ is selected according to a policy (which is to be learned by NSUBS) to be one of the local candidate nodes (line 8), that can be mapped to $u_t$. It is noteworthy that this local candidate node set "$\mathcal{A}_{u_t} \subseteq V_G$" is refined over the global candidate sets $C$ based on the current search state $s_t$. $s_t$ is defined as $(q, G)$ along with the current mapping $M$ and $u_t$. $\mathcal{A}_{u_t} \subseteq V_G$ ensures any node in $\mathcal{A}_{u_t}$ would lead to the extended subgraphs at $s_t$ still being isomorphic to each other. Thus, the local candidate set $\mathcal{A}_{u_t}$ for $u_t$ is the action space, for which we learn a policy (line 9) to order the nodes, resulting in an ordered list $\mathcal{A}_{u_t, \text{ordered}}$.

Despite existing efforts to compute a small $C$, a good $\phi$, and a small $\mathcal{A}_{u_t}$ to reduce search space, we observe that the size of $\mathcal{A}_{u_t}$ can be up to thousands of nodes for many real-world large target graphs, calling for a smarter policy to order the nodes not only in $q$ but also in $G$ ($\mathcal{A}_{u_t}$ to be specific). To the best of our knowledge, all the existing methods adopt a random ordering in $\mathcal{A}_{u_t, \text{ordered}}$. We conjecture it is because existing local candidate computation techniques can further prune nodes from $C$, and thus enumerating $\mathcal{A}_{u_t}$ in a random ordering can be tractable on small graph pairs. We will experimentally show that this attributes to their failure to find a match for many large graph pairs. In fact, a theoretically perfect policy, $policy^*$, can find the entire match of the query graph in $V_q$ steps or recursive calls, assuming $q$ has at least one match with $G$. This again inspires our proposed method to improve the policy for node selection from $\mathcal{A}_{u_t}$.

---

**Algorithm 2** $\texttt{Backtracking}(q, G, C, \phi, M)$

1: **Input:** $q$, $G$, $C$, $\phi$, and current mapping $M$.
2: **Output:** Subgraph match mappings.
3: **if** $|M| = |V_q|$ **then**
4:     output $M$;
5:     return;
6: **end if**
7: $u_t \in V_q \leftarrow \phi(M)$;
8: $\mathcal{A}_{u_t} \leftarrow s_t.getLocalCand(u_t)$;
9: $\mathcal{A}_{u_t, \text{ordered}} \leftarrow policy(s_t, \mathcal{A}_{u_t})$;
10: **for** $v_t$ in $\mathcal{A}_{u_t, \text{ordered}}$ **do**
11:     $M \leftarrow M.add(u_t, v_t)$;
12:     $\texttt{Backtracking}(q, G, C, \phi, M)$;
13:     $M \leftarrow M.remove(u_t, v_t)$;
14: **end for**

---

## 3 PROPOSED METHOD

In this section we formulate the problem of Subgraph Matching as learning an RL agent that grows the extracted subgraphs by adding new node pairs to the current subgraphs. We first describe the environment setup, then depict our proposed encoder-decoder neural network which provides actions for our agent to grow the subgraphs in a search context.

### 3.1 RL FORMULATION OF SUBGRAPH MATCHING

We formulate the Subgraph Matching problem as a Markov Decision Process (MDP), where the RL agent starts with an empty solution, and iteratively matches one node pair at a time until no more nodes can be matched. As mentioned in Section 2.3, our policy assigns a score to each action in $\mathcal{A}_{u_t}$, and therefore can be modeled as a policy network $P_\theta(a_t|s_t)$ that computes a probability distribution over $\mathcal{A}_{u_t}$ for the current state $s_t$, where the node pair to select consists of $(u_t, v_t)$. However, since $u_t$ is chosen by $\phi$, we regard an action as a node $v_t$ selected from $\mathcal{A}_{u_t}$. Since each action is a target node in $\mathcal{A}_{u_t}$, the neural network must learn good node embeddings that capture the current matching status in each search state, which will be shown in Section 3.2. We define our reward as $R^{(+)} = 1$ if the subgraph is fully matched and $R^{(-)} = 0$ otherwise.

To provide ample training signals to the neural network model, during training, we randomly sample subgraphs from the target graph and record the mapping between the nodes in the sampled subgraph and $V_G$. Therefore, after the search, we collect the positive training signals at each state as all the node-node pairs $(u, v)$ that lead to a solution. The collection of such positive pairs is denoted as $P(s_t)$. Negative pairs $N(s_t)$ can be obtained by randomly sampling node-node pairs at each $s_t$ from the action space $\mathcal{A}_{u_t}$. However, the solution sparsity issue mentioned previously causes insufficient positive pairs. We therefore propose the following look-ahead loss function which augments $P(s_t)$ with the positive node-node pairs in future states. Overall, the look-ahead loss function can be

formulated as

$$L_{\text{la}}(s_t) = \sum_{k=0}^{T-t} L_{\text{ce}}(s_{t+k}) \tag{1}$$

where $T$ corresponds to the final step that matches the entire subgraph, and $L'_{\text{la}}$ is the cross-entropy loss computed at $s_{t+k}$. Specifically, $L_{\text{ce}}(s_{t+k})$ is the cross entropy loss and computed as the negative of the following:

$$\sum_{(u,v)\in P(s_{t+k})} R^{(+)} \log \pi_\theta \big((u,v)|s_t\big) + (1 - R^{(+)}) \log \Big(1 - \pi_\theta\big((u,v)|s_t\big)\Big) +$$
$$\sum_{(u,v)\in N(s_{t+k})} R^{(-)} \log \pi_\theta \big((u,v)|s_t\big) + (1 - R^{(-)}) \log \Big(1 - \pi_\theta\big((u,v)|s_t\big)\Big) \tag{2}$$

where $\pi_\theta\big((u,v)|s_t\big) = \sigma\Big(P_{\text{logit}}\big((u,v)|s_t\big)\Big)$ denotes the raw output of the policy network followed by applying the Sigmoid function. Since we define $R^{(+)} = 1$ and $R^{(-)} = 0$, $L_{\text{ce}}(s_{t+k})$ is simplified as

$$-\sum_{(u,v)\in P(s_{t+k})} \log \pi_\theta\big((u,v)|s_t\big) - \sum_{(u,v)\in N(s_{t+k})} \log \Big(1 - \pi_\theta\big((u,v)|s_t\big)\Big) \tag{3}$$

In addition to the look-ahead loss that encourages the policy network to recognize positive node-node pairs, we also adopt the max margin loss, $L_{\text{mm}}$, proposed in Lou et al. (2020) to encourage the encoder to produce subgraph-graph relation aware node embeddings $\boldsymbol{h}$. Our overall loss function is the combination of the novel look-ahead loss function and the max margin loss function, i.e. $L_{\text{total}} = L_{\text{la}} + L_{\text{mm}}$. More details can be found in the appendix.

### 3.2 ENCODER-DECODER DESIGN FOR POLICY ESTIMATION

Our neural network consists of an encoder which produces node-level embeddings for $V_q$ and $V_G$ and a decoder which transforms the node embeddings into $P_\theta(a_t|s_t)$. We identify the following challenges: (1) Each state $s_t$ consists of $q$, $G$, and the mapping $M_t$ between the matched nodes $S_{t,q} \subseteq V_q$ and $S_{t,G} \subseteq V_G$, which should be effectively represented and utilized in predicting $P_\theta(a_t|s_t)$; (2) $P_\theta(a_t|s_t)$ is dependent on $s_t$ and $a_t$, potentially requiring the node embeddings to be recomputed at every search step and incurring too much computational overhead; (3) Existing Graph Neural Networks (GNN) relying on message passing (e.g. Kipf & Welling (2016); Velickovic et al. (2018); You et al. (2020)) are inherently local, which are incompatible with the policy estimation requiring global matching of $q$ to $G$. Even the more recent GNNs that enhance expressiveness beyond 1-Weisfeiler-Lehman test (Morris et al., 2020; Wijesinghe & Wang, 2022) still cannot guarantee the capturing of nodes in the solution, since otherwise the NP-hard task would then be solved. This motivates the following design leveraging the properties of Subgraph Matching as much as possible.

#### 3.2.1 ENCODER: EMBEDDING GENERATION VIA QC-SGMNN

The nodes of $q$ and $G$ are encoded into initial one-hot embeddings $\boldsymbol{h}^{(0)} \in \mathbb{R}^{|\Sigma|}$, and fed into $K$ sequentially stacked Query-Conditioned Subgraph Matching Neural Networks (QC-SGMNN) layers to obtain $\boldsymbol{h}^{(K)} \in \mathbb{R}^D$, where $D$ is the dimension of embeddings, followed by a JUMPING KNOWLEDGE (Xu et al., 2018) network to combine the node embeddings from the $K$ layers to obtain the final node embeddings, denoted as "Aggregate" in Figure 5.

We start by observing that Subgraph Matching desires all nodes in $q$ to be matched with $G$, and existing works in Subgraph Matching aiming at reduce $\mathcal{A}_{u_t}$ (Section 2.3) guarantee no false pruning. In other words, $\mathcal{A}_{u_t}$ is guaranteed to contain the solution nodes in $G$ that should match with $u_t$, that is, $\mathcal{A}_{u_t}$ provides useful hints of which nodes should be matched together. We then compute the candidate space $\mathcal{A}_{u'}$ for every remaining query graph node $u' \notin S_{t,q}$, resulting in a mapping $M'_t = \{u' \mapsto \mathcal{A}_{u'}, \forall u' \in V_q \setminus S_{t,q}\}$. Intuitively, $M'_t$ can be regarded as hints of *future* node-node mappings. In contrast, $M_t = \{u \mapsto \{v\}, \forall u \in S_{t,q}\}$ reflects the *current* state $s_t$, which maps each node in $S_{t,q}$ to a unique node in $G$. We define $\tilde{M}_t$ as the union of $M_t$ and $M'_t$, and define $\tilde{M}_t^{-1}$ as the reverse mapping of $\tilde{M}_t$, which serve as the basis for the matching module.

QC-SGMNN consists of a propagation module, which performs regular intra-graph message passing on $q$ and $G$ individually, and a matching module, which performs state-dependent subgraph-graph

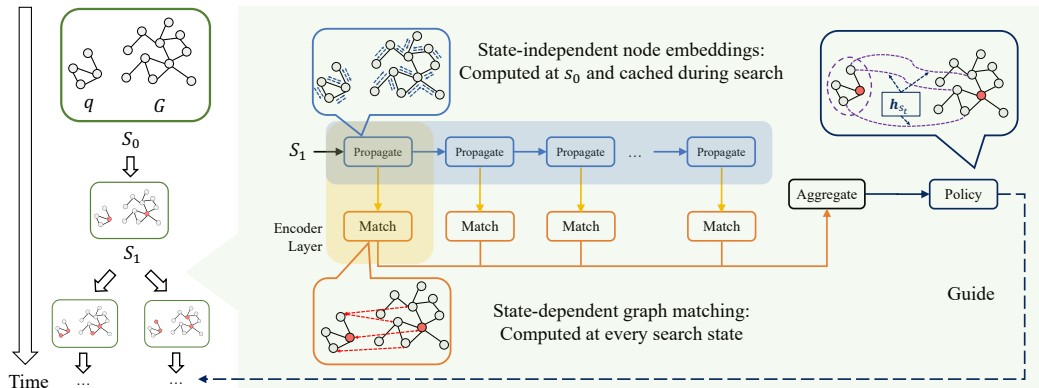

Figure 1: The overall process of Subgraph Matching is a search algorithm that matches one node pair at a time for the input query $q$ and target graph $G$ guided by a learned policy. Due to the large action space incurred by the large $G$ in practice, we propose to train a policy network to guide the selection of local candidate nodes in $G$ at each search state. This requires effective node embeddings to be learned that can reflect the node-to-node mapping at the current state and contribute to the prediction of the policy and reward. To achieve this end, we propose a novel QC-SGMNN encoder layer consisting of a "Propagation" module that performs intra-graph message passing as a typical GNN and a "Matching" module that performs state-dependent inter-graph matching (Section 3.2.1). The $P_\theta(a_t|s_t)$ is trained using the novel loss functions described in Section 3.1.

matching leveraging $\tilde{M}_t$ to pass information between $q$ and $G$. Specifically,

$$
\begin{aligned}
\boldsymbol{h}_{u,\mathrm{intra}}^{(k+1)} &= f_{\mathrm{agg}}\big(\{f_{\mathrm{msg}}(\boldsymbol{h}_{u,\mathrm{intra}}^{(k)}, \boldsymbol{h}_{u',\mathrm{intra}}^{(k)})|u' \in \mathcal{N}(u)\}\big), \\
\boldsymbol{h}_{v,\mathrm{intra}}^{(k+1)} &= f_{\mathrm{agg}}\big(\{f_{\mathrm{msg}}(\boldsymbol{h}_{v,\mathrm{intra}}^{(k)}, \boldsymbol{h}_{v',\mathrm{intra}}^{(k)})|v' \in \mathcal{N}(v)\}\big), \\
\boldsymbol{h}_{q,\mathrm{intra}}^{(k+1)} &= f_{\mathrm{readout}}(\{\boldsymbol{h}_{u,\mathrm{intra}}^{(k+1)}|u \in V_q\}), \\
\boldsymbol{h}_{u,G\to q}^{(k+1)} &= f_{\mathrm{agg}}\big(\{f_{\mathrm{msg}}(\boldsymbol{h}_{u,\mathrm{intra}}^{(k+1)}, \boldsymbol{h}_{v,\mathrm{intra}}^{(k+1)})|v \in \tilde{M}_t(u)\}\big), \\
\boldsymbol{h}_{v,q\to G}^{(k+1)} &= f_{\mathrm{agg}}\big(\{f_{\mathrm{msg}}(\boldsymbol{h}_{v,\mathrm{intra}}^{(k+1)}, \boldsymbol{h}_{u,\mathrm{intra}}^{(k+1)}, \boldsymbol{h}_{q,\mathrm{intra}}^{(k+1)})|u \in \tilde{M}_t^{-1}(v)\}\big), \\
\boldsymbol{h}_{u}^{(k+1)} &= f_{\mathrm{combine}}(\boldsymbol{h}_{u,G\to q}^{(k+1)}, \boldsymbol{h}_{u,\mathrm{intra}}^{(k+1)}), \\
\boldsymbol{h}_{v}^{(k+1)} &= f_{\mathrm{combine}}(\boldsymbol{h}_{v,q\to G}^{(k+1)}, \boldsymbol{h}_{v,\mathrm{intra}}^{(k+1)}).
\end{aligned}
\tag{4}
$$

The first two steps can be any intra-graph message passing GNNs such as Graph Attention Networks (Velickovic et al., 2018) with a message function $f_{\mathrm{msg}}$ and an aggregation function $f_{\mathrm{agg}}$, corresponding to the propagation module. The middle three steps compute intermediate embeddings that will be used for the last two steps, i.e. the matching module. Specifically, $\boldsymbol{h}_{u,G\to q}^{(k+1)}$ and $\boldsymbol{h}_{v,q\to G}^{(k+1)}$ compute the cross-graph message passing from $G$ to $q$ and $q$ to $G$ using $\tilde{M}_t$ and $\tilde{M}_t^{-1}$, respectively (represented as the red dashed lines in Figure 5). A graph-level embedding $\boldsymbol{h}_{q,\mathrm{intra}}^{(k)}$ is computed via $f_{\mathrm{readout}}$ and used in the information passing from $q$ to $G$ to let the embeddings of $V_G$ query-conditioned. We do not inject the graph-level embeddings of $G$ into $\boldsymbol{h}_{u,G\to q}^{(k+1)}$, since the large size of $G$ could result in too much information loss in the readout operation. The last two steps combine the intra-graph embeddings and inter-graph embeddings via $f_{\mathrm{combine}}$ to produce the final output embeddings. It is noteworthy that $f_{\mathrm{msg}}$ and $f_{\mathrm{agg}}$ refer to the general class of functions that yield messages between two nodes and performs aggregation on a set of messages, and in practice, the propagation and matching modules can use different functions for $f_{\mathrm{msg}}$ and $f_{\mathrm{agg}}$.

To address challenge (2), we make the observation that $\boldsymbol{h}_{\mathrm{intra}}^{(k)}$ only depends on $q$ and $G$ and is independent of $M_t$, and thus can be computed once and cached at the beginning of the search (denoted as the top branch in Figure 5) and later reused throughout the search. Our QC-SGMNN decouples the propagation and matching steps and outputs two sources of information separately, allowing the search to cache the state-independent node embeddings and dynamically select the computational paths during search. Thanks to the caching, only the initial iteration requires the $\mathcal{O}(|E_q|+|E_G|)$ computation, and all the subsequent iterations only involve $\mathcal{O}(|V_q||\bar{\mathcal{A}}_{u_t}|)$ complexity, where $|\bar{\mathcal{A}}_{u_t}|$ is the average size of local candidate space.

### 3.2.2 DECODER: $P_\theta(a_t|s_t)$ ESTIMATION

Since the node embeddings of $q$ has received the right amount of information from $\tilde{M}_t$, we propose an attention-based mechanism to compute the state embedding: $\boldsymbol{h}_{s_t} = \sum_{u \in V_q} f_{\text{att}}(\boldsymbol{h}_u, \{\boldsymbol{h}'_u | u' \in V_q\}) \boldsymbol{h}_u$, where $f_{\text{att}}$ computes one attention score per node normalized across $V_q$ to tackle challenge (3). Intuitively, the attention function learns which nodes are important for contributing to the eventual subgraph selected by $s_T$. Due to the cross-graph communication in QC-SGMNN, we only aggregate nodes from $V_q$ to obtain the state representation $\boldsymbol{h}_{s_t}$, taking advantage of the fact that $|V_q|$ is typically much smaller than $|V_G|$ in Subgraph Matching, further addressing challenge (2).

For the policy, we aim to tackle challenge (3), by using $\boldsymbol{h}_{s_t}$. The reasons are two-fold. First, by definition $P_\theta(a_t|s_t)$ requires $s_t$ as input; Second, the attention mechanism used to compute $\boldsymbol{h}_{s_t}$ capturing the future subgraph. Combined with a bilinear tensor product with learnable parameter $\boldsymbol{W}^{[1:F]} \in \mathbb{R}^{D \times D \times F}$ with a hyperparameter $F$ to allow the action node embeddings $\boldsymbol{h}_{u_t}$ and $\boldsymbol{h}_{v_t}$ to fully interact, we obtain $P_{\text{logit}}(a_t|s_t) = \text{MLP}\big(\text{CONCAT}(\boldsymbol{h}_{u_t}^T \boldsymbol{W}^{[1:F]} \boldsymbol{h}_{v_t}, \boldsymbol{h}_{s_t})\big)$, followed by a softmax normalization over the logits for all the actions. The decoder has time complexity $\mathcal{O}\big(|V_q| + |\bar{\mathcal{A}}_{u_t}|\big)$.

## 4 EXPERIMENTS

We evaluate NSUBS against seven backtracking-based algorithms for exact Subgraph Matching, and conduct experiments on six real-word target graphs from various domains, whose details can be found in the supplementary material. We find NSUBS written in Python can outperform the state-of-the-art solver under most cases, suggesting the effectiveness of the learned policy for ordering nodes in $G$. Code, trained model, and all the datasets used in the experiments are released as part of the supplementary material for reproducibility. More results and the ablation study can be found in the supplementary material.

### 4.1 DATASETS AND EVALUATION PROTOCOL

We use one synthetic dataset BA and six real-world target graphs. For DBLP and YOUTUBE, we prepare two query sets of small and large query graphs, denoted as "-S" and "-L", respectively, i.e. DBLP-S, DBLP-L, and YT-S, YT-L. As shown in Table 1, the largest target graph, YOUTUBE, has over 1M nodes and 2M edges.

To train our model, we sample query graphs of any sizes and let NSUBS perform the search algorithm and collect training data from the search tree. Therefore, we sample query graphs from each target graph of sizes ranging from 4 nodes to 128 nodes, and validate that no query graph in the testing set is visible to the model, that is, no

Table 1: Target graph description.

| Dataset | Domain | $|V_G|$ | $|E_G|$ |
|---|---|---|---|
| BA | Synthetic | 10,000 | 29,991 |
| BRAIN | Biology | 1,076 | 90,811 |
| HPRD | Biology | 9,045 | 34,853 |
| COSMOS | Astronomy | 21,596 | 176,830 |
| DBLP | Citation | 317,080 | 1,049,866 |
| ETHE | Financial | 709,351 | 946,582 |
| YOUTUBE | Social | 1,134,890 | 2,987,624 |

graph in the training set is isomorphic to any graphs in the testing set. During the testing stage, for each target graph, we use the following evaluation protocol. For each graph pair, we set a time limit of 5 minutes, and record the result of whether a solution is found or not, and the number of solutions for solved pairs.

### 4.2 BASELINES AND PARAMETER SETTINGS

We compare NSUBS against a series of baseline solvers whose source codes are provided by (Sun & Luo, 2020): LDF (Sun & Luo, 2020), NLF (Sun & Luo, 2020), QUICKSI (Shang et al., 2008), GRAPHQL (He & Singh, 2008), TSO (Han et al., 2013), CFL (Bi et al., 2016), CECI (Bhattarai et al., 2019), and DP-ISO (Han et al., 2019). We also include NMATCH (Lou et al., 2020) as a learning-based method for exact Subgraph Matching, by adapting the original implementation provided by the authors of (Lou et al., 2020) into our backtracking search framework. Specifically, at each search step, we invoke the inference step of the released trained NMATCH model over the input graphs, and obtain a matching score for each $(u_t, v_t)$ action node pair to guide the selection of nodes in $\mathcal{A}_{u_t}$.

Since we are among the first to learn a policy for selecting nodes in $G$, we use DP-ISO for filtering and GRAPHQL for query node ordering. We implement the backtracking search framework and DP-ISO's local candidate computation algorithm in Python.

NSUBS uses 8 layers of the proposed QC-SGMNN encoders. $\omega$ is set to 10 and max_iters is set to 120. Training is performed on a server with NVIDIA Tesla V100 GPUs. We train NSUBS for approximately 2 days with randomly sampled subgraphs for each target graph with the AdamW optimizer (Loshchilov & Hutter, 2017) with the initial learning rate 0.0005 and the $\epsilon$ parameter set to 0.01 for numerical stability. Gradient clipping is applied with the maximum norm of the gradients set to 0.1. We use a validation set to select the best model to evaluate.

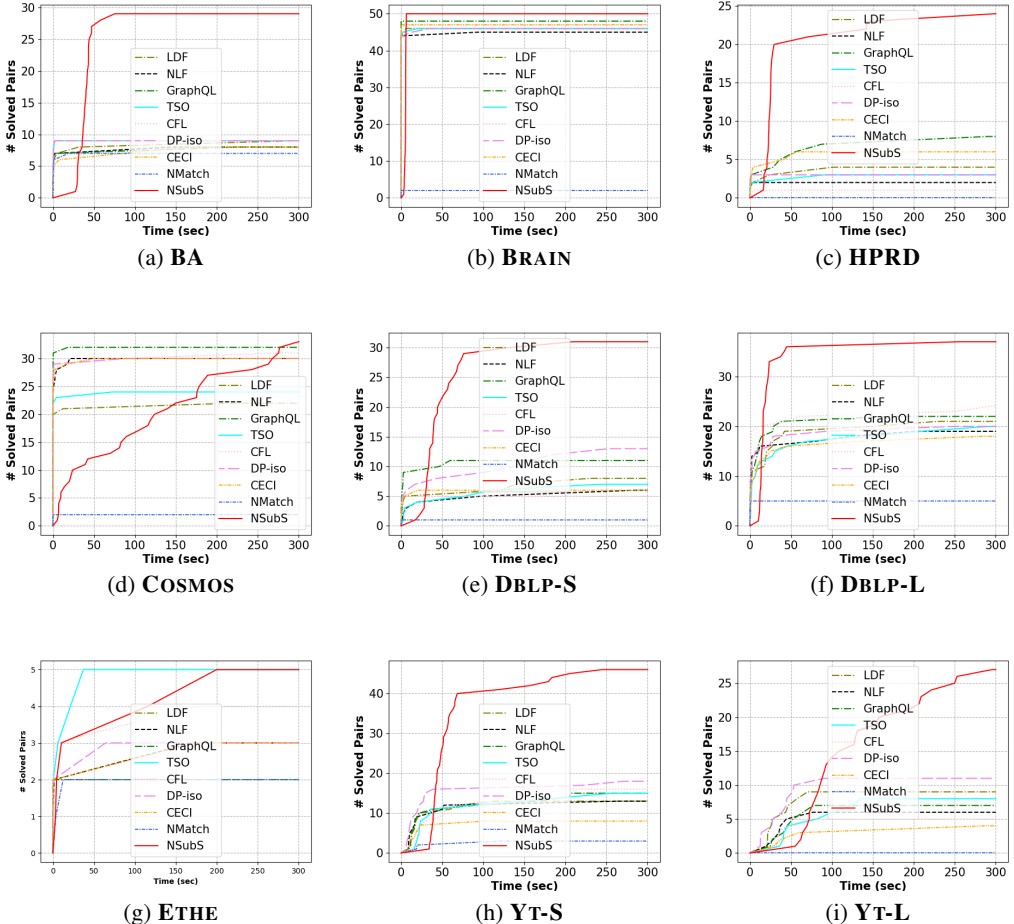

Figure 2: The growth of the number of solved pairs across time.

## 4.3 RESULTS

To examine the efficiency of each method and analyze the efficacy across time, we conduct the following evaluation. For each $(q, G)$ pair, we record the time the method takes to find a solution, and accumulate the number of solved pairs across time. From $t = 0$ to $t = 300$ seconds, an increase at $t$ indicates the method solves one graph pair at $t$. The earlier a method solves the graph pairs, the faster and better the method is.

In theory, given an infinite amount of time, every method adopting the backtracking search algorithm would be able to solve a graph pair. However, such assumption is not practical. Thus, the curves in Figure 2 have the practical implication that under a reasonable amount of time budget, the idea of using a smarter policy indeed brings better performance to Subgraph Matching. Another observation is that the baseline solvers flatten towards the end of 5 minutes, indicating that they get stuck in

Table 2: Number of solutions found by each method after 5 minutes averaged by the number of the graph pairs in each dataset. For clarity and compactness, each result has been divided by 1000, i.e. each number is in the unit of $10^3$.

| Method | BA | BRAIN | HPRD | COSMOS | DBLP-S | DBLP-L | ETHE | YT-S | YT-L |
|---|---|---|---|---|---|---|---|---|---|
| LDF | 1.21 | 2.05 | 0.69 | 4.80 | 4.82 | 1.51 | 2.50 | 2.73 | 1.75 |
| NLF | 1.29 | 2.24 | 0.37 | 7.05 | 4.26 | 1.06 | 2.34 | 2.52 | 1.06 |
| GRAPHQL | 1.42 | 3.95 | 1.24 | **7.70** | 5.23 | 2.55 | **3.53** | 2.73 | 1.23 |
| TSO | 1.41 | 1.26 | 0.50 | 5.53 | 4.36 | 1.36 | 2.30 | 2.57 | 1.24 |
| CFL | 1.56 | 1.12 | 0.18 | 6.97 | 5.70 | 1.13 | 2.32 | 2.84 | 0.90 |
| DP-ISO | 1.31 | 2.11 | 0.54 | 6.93 | 4.86 | 2.63 | 2.58 | 3.32 | 1.88 |
| CECI | 1.21 | 1.03 | 1.05 | 6.96 | 4.18 | 1.35 | 1.70 | 1.60 | 0.55 |
| NMATCH | 1.02 | 0.00 | 0.00 | 0.24 | 1.25 | 0.24 | 1.39 | 0.37 | 0.00 |
| NSUBS | **2.37** | **4.58** | **4.17** | 3.70 | **8.32** | **6.24** | 2.53 | **7.93** | **3.62** |

unpromising search states that are unlikely to contain the solution, confirming the severity of the aforementioned challenges of solution and reward sparsity.

As shown in Figure 2, NSUBS initially seems not as good as other models in the first few seconds due to the additional neural network computational overhead at each step, but achieves the same or better performance on all of the 9 query sets after 5 minutes. We observe baseline solvers tend to fail on larger target graphs, as search space pruning on its own is not effective enough to guarantee solutions in such cases with a large amount of candidate nodes. Learning-based methods outperform solver baselines because they provide a better target graph node ordering, under the same search framework.

Out of the learning-based methods, NSUBS consistently outperforms NMATCH, as it features a more powerful encoder that efficiently computes a policy conditioned on the current search state with a better inter-graph communication mechanism. This also suggests that the target graph node ordering provided by NSUBS is much better than the random ordering adopted by solver baselines, confirming that our novelties indeed greatly improve Subgraph Matching performance.

We observe that when the query graph size increase, all methods tend to show lower performance, which can be attributed to the exponentially growing search space. It is noteworthy that the survey paper comparing existing solvers (Sun & Luo, 2020) uses query graphs up to 32 nodes, whereas we challenge all methods by testing on query graphs up to hundreds of nodes. The fact that NSUBS is able to solve more graph pairs than baselines when the target graphs are large demonstrates good scalability of NSUBS.

Given the nature of Subgraph Matching, we also evaluate the ability of each method to find as many solutions as possible. As Table 2 indicates, NSUBS outperforms baseline methods on 7 out of 9 datasets, suggesting that the slower but more intelligent NSUBS can not only ensure one solution is found, but also provide as many as and even more solutions compared against the faster baseline solvers. The exceptions are HPRD and ETHE, where GRAPHQL solves less pairs as shown in Figure 2, but on average finds more solutions per graph pair across these two datasets. We empirically observe that the baseline solvers tend to find many solutions because as soon as one solution is found, nearby solutions can be found. In contrast, NSUBS continues performing neural network operations after one solution is found. This indicates the efficiency limitation of the current model, and calls for the need for future efforts to speed up the neural network computation, design even better policy network and training methods to allow discovery of more solutions in lesser iterations, etc.

## 5 CONCLUSION

In this paper, we tackle the challenging and important task of Subgraph Matching, and present a new method for efficient and effective exact Subgraph Matching. It contains a novel encoder-decoder neural network architecture trained using a novel look-ahead loss function and a max margin loss to improve the policy estimation as well as the encoder. The core component, QC-SGMNN, is a query-conditioned graph encoder that performs intra-graph propagation and inter-graph node matching to capture each state. To address the reward sparsity issue posed by the large action space, we augment the positive node-node pairs at each search step with future states' node-node pairs. We experimentally show the utility of the proposed NSUBS method on the important Subgraph Matching task. Specifically, NSUBS is able to solve more graph pairs than several existing Subgraph Matching solvers on one synthetic dataset and six large real-world datasets.

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

## A  DETAILS ON TRAINING AND TESTING OF NSUBS

### A.1  DETAILS ON COLLECTING TRAINING SIGNALS

As mentioned in the main text, we utilize the max margin loss proposed in Lou et al. (2020) which can be written as

$$L_{\mathrm{mm}}(s_{t+k}) = \sum_{(u,v)\in P(s_{t+k})} E(\boldsymbol{h}_u, \boldsymbol{h}_v) + \sum_{(u,v)\in N(s_{t+k})} \max\{0, \alpha - E(\boldsymbol{h}_u, \boldsymbol{h}_v)\}$$

where

$$E(\boldsymbol{h}_u, \boldsymbol{h}_v) = ||\max\{0, \boldsymbol{h}_u - \boldsymbol{h}_v\}||_2^2.$$

The intuition behind the loss function is to encourage the encoder to produce node embeddings that capture the subgraph-graph relationship. Specifically, the loss encourages each dimension of $\boldsymbol{h}_u$ to be less than $\boldsymbol{h}_v$, and the error term $E$ measures the deviation of the current node embeddings from this desired property.

It is noteworthy that both $L_{\mathrm{la}}$ and $L_{\mathrm{mm}}$ rely on the positive and negative node-node pairs, which are obtained by performing search on $(q, G)$ pairs where $q$ is the randomly sampled training subgraph queries from $G$. Therefore, the overall NSUBS model does not need any pre-solved graph pairs for training.

Here we describe more details on how we prepare the positive node-node pairs $P(s_t)$. During training, we set a time limit of 5 minutes for each $(q, G)$ pair, and at the end of the search process, we obtain a search tree where each node in the search tree corresponds to a state $s_t$. For a state $s_t$, we look ahead at all sequences of states leading to a solution starting from $s_t$, and compute the $L_{\mathrm{la}}$ loss according to Equation 1 for each sequence, and sum up the loss for all such sequences. To obtain $N(s_t)$, we randomly sample $|P(s_t)|$ node-node pairs from the action space $\mathcal{A}_{u_t}$, where $u_t$ is the query graph node selected by $\phi$ as described in Section 2.3, and $\mathcal{A}_{u_t}$ is the local candidate nodes in $G$ to be selected by the neural network.

We provide an illustration of the collection of positive node-node pairs in Figure 3. In this example, there are two sequences of states leading to a solution of the full match of $q$ to $G$, for which we label the action node pair associated with each action. For clarity, we only show the first few states and omit the later states. It is noteworthy that search states that do not lead to a solution, e.g. $s_1$, do not produce any training signal, since there is insufficient information regarding such states. Since we always perform random subgraph sampling to generate training graph pairs, we ensure at least one sequence of actions leading to a solution by forcing the initial search trajectory to follow the ground-truth node-node mapping obtained when conducting sampling. In Figure 3, this corresponds to the initial trajectory (0,0), (1,2), (2,5), (3,9), (4,7) . . .

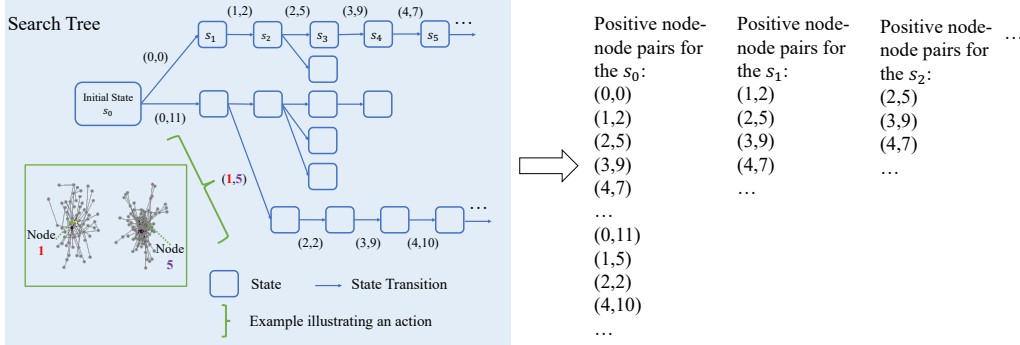

Figure 3: The search tree after the 5-minute time budget provides positive training signals. An action node pair $(u_t, v_t)$ where $u_t \in V_q$ and $v_t \in V_G$ is associated with an edge. Negative training pairs are randomly sampled from $V_q \times V_G$.

A.2  DETAILS ON TESTING GRAPH PAIRS

We conduct evaluation on the following testing graph datasets:

1. The synthetic dataset BA is unlabeled and generated using the Barabási–Albert model (Barabási & Albert, 1999) with the parameter of attachment set to 3. We sample 50 query graphs each of 64 nodes as the query set. The sampling algorithm starts with a randomly selected initial node, and then performs random walk until a subgraph consisting of a pre-defined number of nodes is sampled. Each step in the random walk selects a node from the neighboring nodes of the current node to visit in the next step, where the node can be a new node or an already selected node, i.e. the random walk is allowed to backtrack. We

define the raw sampling probability to be $1/p$ if the node is the selected subgraph, and $p$ otherwise. A small $p$ therefore encourages re-visitation and the final subgraph tends to be "star"-like, while a large $p$ typically yields a more "path"-like subgraph. Each of the 50 sampled subgraphs uses a different $p$ parameter by sweeping across the range from 0.001 to 1000. Specifically, the $i$-th sampled subgraph uses $p = 0.001 * \exp\left(\log(10^6)/49 * (i - 1)\right)$, i.e. the first subgraph uses $p = 0.001$, the second uses $p = 0.00133$, the third uses $p = 0.00176$, etc. The 49th subgraph uses $p = 754.312$, and the last uses $p = 1000$. Without specific mentioning, the sampling of subgraphs for other datasets uses this algorithm.

2. The BRAIN dataset is obtained from mouse retina connectome (Helmstaedter et al., 2013) with node labels representing the coarse type determined from supplemental text (Jonas, 2015). We sample 50 query graphs each of of 64 nodes. There are 5 types of node labels in total.

3. The HPRD target graph is a protein-protein interaction graph obtained from a recent survey paper on Subgraph Matching (Sun & Luo, 2020) without node labels. Similar to BRAIN, we sample 50 64-node graphs as the query graphs.

4. The COSMOS graph represents the network of galaxies (Coutinho et al., 2016) without node labels, for which we sample 50 query graphs each of 32 nodes as queries.

5. The DBLP and YOUTUBE target graphs are also used in the survey paper (Sun & Luo, 2020), but instead of sampling subgraphs as query graphs, we rely on ground-truth communities mined from an algorithm (Yang & Leskovec, 2012) as query graphs. According to Leskovec & Krevl (2014), for DBLP, authors who published to a certain journal or conference form a community, and for YOUTUBE, user-defined groups are considered as ground-truth communities. Since many ground-truth communities are provided by Leskovec & Krevl (2014), we sample a 50-graph small query set and a 50-graph larger query set for these two datasets. Specifically, DBLP-S contains communities of 20-30 nodes, DBLP-L contains communities of 30-60 nodes, YT-S contains communities of 30-40 nodes, and YT-L contains communities of 40-100 nodes.

6. The ETHE dataset is obtained from a recent paper on financial anomoly detection (Chen & Tsourakakis, 2022). The graph represents 1-week Ethereum blockchain transactions where each edge represents a transaction with the amount. We run the released code provided by the paper to extract top anomalous subgraphs, and filtering subgraphs that are too small or too large, yielding 24 subgraphs ranging from 6 to 822 nodes as query graphs. Intuitively, these subgraphs are relatively dense and correlate with possibly malicious behavior. We remove the transaction amount from the edges, and run Subgraph Matching using the mined subgraphs as query graphs and the original transaction graph as the target graph. Although NSUBS can potentially be extended for clique or anomaly detection, we leave the investigation as future work.

## A.3 DETAILS ON TRAINING AND VALIDATION GRAPH PAIRS

For each target graph, we sample query graphs of various sizes to perform curriculum learning, as shown effective by earlier graph matching works (Bai et al., 2021; Lou et al., 2020). We ensure the testing query graphs and training query graphs are different by using different random seeds and further running an isomorphism checker to ensure none of the testing query graphs are visible during training. The query graphs come in the following sizes: 8, 16, 24, 32, 48, 64, 96, and 128 nodes. It is noteworthy that the query graphs are obtained from provided ground-truth communities or mined subgraphs for DBLP, ETHE and YOUTUBE for testing, but during training, all the query graphs are obtained via subgraph sampling.

Regarding the validation scheme, we use the following validation set for validating the models: 3 query graphs of size 8, 3 query graphs of size 16, 3 query graphs of size 32, 3 query graphs of size 64, and 3 query graphs of size 128. These 15 query graphs are sampled from the same target graph with different random seeds, and we ensure they are not isomorphic to any query graphs in the training set (nor isomorphic to any query in the testing set if the testing set is obtained via random subgraph sampling).

The overall training process consists of the following iterative steps: (1) Sample a query graph from $G$ and perform search with a limit of 5 minutes; (2) Collect training signals and add them to a replay

buffer; (3) Perform training by sampling a batch of training examples from the replay buffer and running back-propagation; (4) Once every 5 times of performing search on training graph pairs, we iterate through the 15 graph pairs in the validation set with a time budget of 40 seconds, and record the validation reward, which is defined as the average best subgraph size. We accept the trained model if the validation reward improves, and revert to the previous best model otherwise by loading the best model's checkpoint. At the end of the training process, we evaluate the best model that achieves the best validation performance.

## A.4 DETAILS ON HYPERPARAMETERS

For the encoder, we set the intra-graph $f_{\text{agg}}$ and $f_{\text{msg}}$ functions to GRAPHSAGE (Hamilton et al., 2017), and set the inter-graph $f_{\text{agg}}$ function to be SUM and $f_{\text{msg}}$ function to be dot-product style attention (Vaswani et al., 2017) over incoming messages. Specifically, the message collected by each query node, $\boldsymbol{h}_{u,G\to q} \in \mathbb{R}^D$, consists of attended node embeddings from $v' \in \tilde{M}_t(u)$. Overall, $\boldsymbol{h}_{u,G\to q}$ is computed as

$$\sum_{v' \in \tilde{M}_t(u)} \text{softmax}_{v' \in \tilde{M}_t(u)}\big(\text{MLP}_q(\boldsymbol{h}_{u,\text{intra}})^T \text{MLP}_G(\boldsymbol{h}_{v',\text{intra}})\big)\text{MLP}_{\text{VAL},q}(\boldsymbol{h}_{v',\text{intra}}). \quad (5)$$

The message collected by each target node, $\boldsymbol{h}_{v,q\to G} \in \mathbb{R}^{2D}$, consists of attended node embeddings from $u' \in \tilde{M}_t^{-1}(v)$ and the whole query graph embedding, $\boldsymbol{h}_{q,\text{intra}}$. Overall, $\boldsymbol{h}_{v,q\to G}$ is computed as

$$\boldsymbol{h}_{v,q\to G} = \sum_{u' \in \tilde{M}_t^{-1}(v)} \text{CONCAT}(\boldsymbol{h}'_{v,q\to G}, \boldsymbol{h}_{q,\text{intra}}), \quad (6)$$

where

$$\boldsymbol{h}'_{v,q\to G} = \text{softmax}_{u' \in \tilde{M}_t^{-1}(v)}\big(\text{MLP}_q(\boldsymbol{h}_{u',\text{intra}})^T \text{MLP}_G(\boldsymbol{h}_{v,\text{intra}})\big)\text{MLP}_{\text{VAL},G}(\boldsymbol{h}_{u',\text{intra}}). \quad (7)$$

In total, we use four different MLPs, i.e. $\text{MLP}_q$, $\text{MLP}_G$, $\text{MLP}_{\text{VAL},q}$, and $\text{MLP}_{\text{VAL},G}$ following (Vaswani et al., 2017), with an $\text{ELU}(\cdot)$ activation after the final layer.

We set the $f_{\text{combine}}$ function to be the concatenation CONCAT followed by MLP. We choose MEAN as the readout function $f_{\text{readout}}$ for computing the query graph-level embedding. We choose the MAX function as the aggregation function for our Jumping Knowledge network which aggregates the node embeddings. We apply layer normalization (Ba et al., 2016) to the final node embeddings before feeding into the decoder.

For the encoder, we stack 8 layers of intra-graph message passing with dimension 16. We stack 8layers of the inter-graph message passing with all MLPs outputting dimension 16. For the decoder, we set $\text{MLP}_{\text{att}}$ to dimensions [16, 4, 1] and $\text{MLP}_v$ to dimensions [16, 32, 16, 8, 4, 1]. We set the policy head's bilinear layer, $\boldsymbol{W}^{[1:F]}$, to output dimension $F = 32$ and the policy head's MLP to dimensions [48, 32, 16, 8, 1]. For all the MLPs, we use the $\text{ELU}(x)$ activation function on the hidden layers. For $\text{MLP}_v$, we apply a $\text{LeakyReLU}(\cdot)$ function following the final layer.

The goal of the encoder is to learn good node embeddings that capture the structural matching of $q$ and $G$, and thus we do not encode the node labels as initial encodings. Instead, we rely on the candidate set generation algorithm to handle the constraint that only nodes of the same label can match. Therefore, it is critical to encode the nodes into initial encodings properly. A simple way to encode the nodes is to assign a constant encoding to every node in both $q$ and $G$. To enhance the initial encodings, we adopt Local Degree Profile (LDP) scheme (Cai & Wang, 2018) concatenated with a 2-dimensional one-hot vector indicating whether a node is currently selected or not.

Our replay buffer is of size 128. During the training process, we repeatedly run 5 minutes of search, populating the replay buffer, 3 minutes of replay buffer sampling and back-propagation, training on as many replay buffer samples as possible, and performing validation to choose whether or not we commit the learned model. One iteration of this process typically takes around 10 minutes. Hyperparameters were found by tuning for performance manually against the validation dataset.

## B NOTES ON SEARCH

### B.1 INDUCED VS NON-INDUCED SUBGRAPHS

One key detail regarding the subgraph definition is whether the subgraph is induced or non-induced. An induced subgraph $q[S]$ consists of all the nodes $S \subseteq V_q$ and requies that for every edge $(u, u') \in E_q$, if both $u$ and $u'$ are in $S$, then the edge $(u, u')$ must be included in the edges of $q[S]$ as well. Intuitively, an induced subgraph does not allow any edges in the original graph to be dropped, as long as both endpoints of the edge are included by the subgraph. In contrast, a non-induced subgraph $q[S]$ consists of all the nodes $S \subseteq V_q$ but allows edges in $E_q$ to be dropped. In this paper, we adopt the definition of non-induced subgraph, which is consistent with Han et al. (2019).

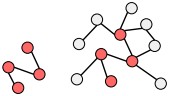 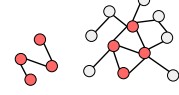

$q$ is an induced subgraph of $G$.     $q$ is a non-induced subgraph of $G$.

Figure 4: Comparison between an induced (left) and a non-induced (right) subgraph. We use the red color to denote the nodes that are included in the match.

## B.2   CHOICE OF FILTERING, QUERY NODE ORDERING, AND LOCAL CANDIDATE COMPUTATION

Motivated by prior work benchmarking solver baselines (Sun & Luo, 2020), we adopt DP-ISO as the filtering method, GRAPHQL as the query node ordering method, and GRAPHQL as the local candidate computation method in NSUBS and NMATCH. As shown by the experimental results in the main paper, this design choice plays a much smaller role in Subgraph Matching than smart target graph node ordering, highlighting the need for NSUBS. Furthermore, we find that optimal filtering, query node ordering, and local candidate computation settings are highly dataset-dependent and hard to determine beforehand; however, because NSUBS's design is orthogonal to such choices, it could easily be adapted to different filtering and query node ordering algorithms. We leave such study to future work.

The filtering method produces a global candidate set, $C : V_q \rightarrow V_t$, to drastically prune the search space of possible matchings. A simple method to compute $C$ is to only allow mappings from $u_t$ to $v_t$ if $v_t$ has degree higher than or equal to the degree of $u_t$. Different filtering algorithms use different such rules (Sun & Luo, 2020). The query node ordering gives a search plan on which $u$ nodes to select first. A simple method is picking the $u$ nodes with the least candidate target nodes given by $C$. Different node ordering algorithms use different such rules (Sun & Luo, 2020).

During the search process, the local candidate set guarantees all subgraph mappings found throughout the search process are isomorphic. Specifically, it ensures $u_t$ is matchable to $v_t$, i.e. $v_t \in \mathcal{A}_{u_t}$ only if (1) all edges between $u_t$ and the currently matched query subgraph (with node set $S$) exist between $v_t$ and the currently matched target subgraph: $(u_t, u') \in E_q, u' \in S \implies (v_t, M(u')) \in E_G$ and (2) the matching, $(u_t, v_t)$ exists in the candidate set: $v_t \in C(u_t)$. There are many different implementations and optimizations to ensure these conditions hold. For instance, a query tree data structure (Han et al., 2019) can be stored, allowing $O(1)$ lookup for $(u_t, u') \in E_q$ connections given a particular query node ordering. Different local candidate computation algorithms use different such optimizations (Sun & Luo, 2020). Even though the subgraph isomorphism constraint is ensured through the local candidate set, Subgraph Matching is by no means an easy task due to the large amount of nodes in $\mathcal{A}_{u_t}$.

Among different techniques to further improve the accuracy of Subgraph Matching approaches, learning a state-dependent policy to select nodes from $q$ is one direction worth mentioning. We currently only learn to select nodes in $\mathcal{A}_{u_t} \subseteq V_G$ partly because there is usually a small amount of nodes in $q$, e.g. 32 nodes or 128 nodes, while the number of nodes in $\mathcal{A}_{u_t}$ is much larger. As mentioned in the main text, RL-QVO (Wang et al., 2022) learns a state-independent node ordering policy, $\phi$, via RL that is executed before search. We call such $\phi$ query node ordering. We argue it is a promising future direction to explore learning to select both nodes in $q$ and nodes in $G$ at each search iteration.

As shown effective by previous works, we adapt a promise-based search strategy (Bai et al., 2021), which backtracks to any earlier search state, instead of the immediate parent, whenever a terminal state is reached. We choose which earlier search state to backtrack to by computing a 2:1 weighted average of the search depth normalized by the query graph size and the percentage of explored actions, allowing search to quickly exit local minima. We include this adaptation on all learning methods. We also tried the promise-based search on the solver baselines, but found the performance difference

is minimal, typically around 2%. This is because solver baselines do not assign any ordering to the target graph nodes, thus there is no clear incentive to backtracking early.

## C  COMPARISON WITH RELATED WORKS

### C.1  COMPARISON WITH GRAPH MATCHING NETWORKS

Although GRAPH MATCHING NETWORKS (Ling et al., 2020) provides soft node-node correspondence scores, it does not provide a discrete subgraph matching. A naive solution is thresholding the soft correspondence matrix to obtain a matching. But thresholding does not guarantee a 1-1 mapping between query graph and target graph nodes. Another idea is to use the hungarian algorithm (Kuhn, 1955) to obtain a 1-1 mapping, but this doesn't guarantee the matched subgraphs would be isomorphic. Hence, a search framework is required to extract the discrete subgraph matching.

We experimentally verify that NSUBS's QC-SGMNN encoder is superior to GRAPH MATCHING NETWORKS's encoder. This is because QC-SGMNN ignores noisy inter-graph messages by filtering them out with the local candidate set, learns unique representations for each state by being search state dependent, and caches intra-graph messages for more efficient execution.

### C.2  COMPARISON WITH DMPNN

Although DMPNN (Yuan et al., 2022) provides soft node scores in the target graph, it does not provide the discrete subgraph matching. A naive solution is thresholding the soft scores to obtain a discrete subset of target graph nodes, but this cannot guarantee the extracted nodes are isomorphic to the query due to the lack of any way to guarantee that. Hence, a search framework is required to extract the discrete subgraph matching.

We experimentally verify that NSUBS's QC-SGMNN encoder is superior to DMPNN's encoder in Section D.1.2. This is because QC-SGMNN is state-dependent and can learn unique representations for each search state. Furthermore, QC-SGMNN performs interg-raph message passing, that DMPNN misses.

### C.3  COMPARISON WITH MAXIMUM COMMON SUBGRAPH (MCS) DETECTION AN GLSEARCH

#### C.3.1  COMPARISON WITH MCS

Subgraph Matching and MCS detection are highly related tasks, and one can convert Subgraph Matching to MCS detection by feeding the input $(q, G)$ pair to an MCS solver, and check if the MCS between $q$ and $G$ is identical to $q$. However, we note the following differences at the task level:

1. Subgraph Matching requires all the nodes and edges in $q$ to be matched with $G$, whereas MCS detection does not require one of two input graphs to be contained in another.
2. Subgraph Matching by definition requires $q$ to be smaller or equal to $G$, since otherwise the solver can immediately return "no solution". In practice, the query graph is usually given by the user as input, and usually contains less than 100 nodes (Sun & Luo, 2020).

The first difference has several consequences. First, although one can solve Subgraph Matching via an MCS solver, the other way does not hold, i.e. one cannot use a Subgraph Matching solver for MCS detection, since Subgraph Matching has the stronger constraint of matching the entire $q$. Second, the stronger constraint of Subgraph Matching allows existing search algorithms to design pruning techniques as mentioned in the main text.

The second difference implies that the learning models designed for Subgraph Matching may not work well for MCS detection and vice versa. For example, NSUBS has a novel encoder layer with a matching module that has time complexity $\mathcal{O}(|V_q||\bar{\mathcal{A}}_{u_t}|)$. However, if $q$ becomes prohibitively large, e.g. as large as $G$ containing millions of nodes, then the matching step would become a bottleneck and make the overall approach too slow to be useful in practice. Therefore, NSUBS cannot be used for MCS detection not only because fundamentally the search algorithm leverages the stronger constraint if Subgraph Matching, but also because the learning model would be inefficient and thus useless in practice. By the same reasoning, NSUBS is likely to be ineffective for graph isomorphism checking

when both input graphs are of the same size and very large. This suggests the difficulty of designing a general neural network architecture that works efficiently and effectively for a series of related but different combinatorial optimization tasks. In theory, many such tasks are equivalent and can be converted to each other, but in practice, specialized models should be designed to render a truly useful learning-based approach.

### C.3.2 COMPARISON WITH GLSEARCH

Although GLSEARCH 's maximum common subgraph (MCS) framework can theoretically be applied for subgraph matching, in practice, search algorithms designed specifically for subgraph matching work much better than those designed for maximum common subgraph (Archibald et al., 2019). Thus, adaptations are needed to effectively use GLSEARCH for Subgraph Matching.

In terms of RL framework, GLSEARCH uses value networks instead of policy networks, which is not effective for Subgraph Matching. Concretely, GLSEARCH performs the execution of each action to get its next state $s_{t+1}$, and compute the value associated with $s_{t+1}$ for the $Q(s_t, a_t)$. This is named as "factoring out action" in Bai et al. (2021), and requires sequentially going through all the actions to obtain the next states, for which GLSEARCH adopts a heuristic to reduce the amount of actions. There is not such a heuristic in Subgraph Matching, and hence value networks become very inefficient. In contrast, NSUBS efficiently computes one embedding per node for both $q$ and $G$ using the QC-SGMNN encoder layer, and computes $P_\theta(a_t|s_t)$ in the decoder by batching the node embeddings that are in the action space, i.e. $P_{\text{logit}}(a_t|s_t) = \text{MLP}\big(\text{CONCAT}(\boldsymbol{h}_{u_t}^T \boldsymbol{W}^{[1:F]} \boldsymbol{h}_{v_t}, \boldsymbol{h}_{s_t})\big)$ can be turned into a batch operation for all the node embeddings involved in the action space of $s_t$.

In terms of model design, GLSEARCH only performs interaction between the 2 graphs in the decoder, whereas NSUBS interacts the 2 graphs after each graph neural network layer. Because of this, NSUBS decouples the propagation and matching modules to cache the intra-graph embeddings for efficient computation. Furthermore, the query-conditioning in NSUBS's QC-SGMNN captures the asymmetric nature of Subgraph Matching, unlike GLSEARCH's model.

In terms of loss design, GLSEARCH opts for Deep-Q Network loss instead of lookahead loss, because it uses factored out value-networks. This means it does not fully exploit the loss signal. Most importantly, GLSEARCH cannot use the max margin geometric loss term (Lou et al., 2020), since GLSEARCH does not provide node embedding pairs at state $s_t$ to represent action $a_t$.

We experimentally verify that NSUBS's choice of policy network, model design, and loss is superior to GLSEARCH.

### C.4 COMPARISON WITH ISONET

We appreciate ISONET's contribution towards using edge-centric geometric representations (Roy et al., 2022); however, it is not directly comparable because ISONET aims to rank which query graphs are most likely to appear in the target graph, without providing the discrete subgraph matching.

While ISONET provides soft edge-edge and node-node correspondence scores for interpretability, it does not guarantee a discrete sugraph matching. ISONET indeed reports some case studies where the hungarian algorithm can extract some subgraph matchings that satisfies isomorphism constraints, this fails to scale to very large query and target graphs. Thus, a search framework is required to reliably extract the discrete subgraph matching.

Model-wise, ISONET passes each graph through intra-graph message passing then computes the edge-egde or node-node alignment/permutation matrix by taking the dot product of node or edge embeddings from the query and target graphs. NSUBS's QC-SGMNN improves this by introducing an efficient query-conditioned matching module between intra-graph message passing layers. Loss-wise, NSUBS's lookahead loss improves ISONET by merging contrastive learning loss (Lou et al., 2020; Roy et al., 2022) with RL loss.

### C.5 COMPARISON WITH RL-QVO

RL-QVO is orthogonal to NSUBS, proposing a better query vertex-ordering scheme for subgraph matching. For large graphs, target-vertex ordering becomes the bottleneck instead of query vertex-

ordering, as enumerating all target vertices for each query vertex becomes infeasible by the NP-Hard nature of subgraph matching.

Unlike RL-QVO, which uses random target vertex-ordering, NSUBS learns a policy network to perform target vertex-ordering. In addition to RL-QVO 's single graph encoder, NSUBS considers the query to target graph matching with QC-SGMNN. Unlike RL-QVO 's simple PPO loss, NSUBS learns geometric node representations with lookahead loss.

# D   ABLATION STUDIES AND ADDITIONAL RESULTS

## D.1   ABLATION STUDIES

We perform a series of ablation studies whose results are shown in Table 3. We find our key novelties indeed greatly contribute to NSUBS's superior performance on Subgraph Matching, particularly the encoder-decoder design.

### D.1.1   VARIANTS OF NSUBS

Table 3: Ablation study results on Subgraph Matching over HPRD. The ratio of the number of solved pairs over the 50 pairs in the query set is reported. The average number of solutions (divided by $10^3$) is reported too.

| Model | Solved % | # Solutions |
|---|---|---|
| LDF | 0.08 | 0.69 |
| NLF | 0.04 | 0.37 |
| GRAPHQL | 0.16 | 1.24 |
| TSO | 0.06 | 0.50 |
| CFL | 0.02 | 0.18 |
| DP-ISO | 0.06 | 0.54 |
| CECI | 0.12 | 1.05 |
| NMATCH | 0.00 | 0.00 |
| NSUBS | **0.48** | **4.17** |
| NSUBS-rand-neg | 0.36 | 3.35 |
| NSUBS-no-look-ahead | 0.42 | 3.68 |
| NSUBS-no-mm | 0.08 | 0.73 |
| NSUBS-2-encoder-layers | 0.24 | 0.25 |
| NSUBS-$D = 4$ | 0.04 | 0.30 |
| NSUBS-GIN | 0.30 | 2.73 |
| NSUBS-no-Q-matching | 0.34 | 2.98 |
| NSUBS-no-LDP-encode | 0.26 | 2.39 |
| GMN-encoder | 0.30 | 2.89 |
| DMPNN-encoder | 0.32 | 2.25 |
| GLSEARCH-adapted | 0.22 | 1.88 |

Table 3 shows the ablation study results.

1. We first perform the random sampling of negative node-node pairs ($N(s_t)$ in Equation 2) from all the node-node pairs in $V_q$ and $V_G$ instead of from $\mathcal{A}_{u_t}$. We denote the model as "NSUBS-rand-neg", which performs worse than NSUBS but still outperforms baselines. This suggests that the negative sampling should be performed using "harder" examples sampled from the local candidate space instead of all the possible node-node pairs. Such harder negative examples train the model to better identify the promising actions over the unpromising ones at each state.

2. We replace the look-ahead loss with a simpler version, by only looking at the current state's positive node-node pairs without considering future states. NSUBS-no-look-ahead performs worse than NSUBS, indicating the usefulness of the look-ahead loss.

3. We remove the max margin loss, and as a result, NSUBS-no-mm performs much worse, suggesting the necessity for training the encoder to produce good node embeddings. Since

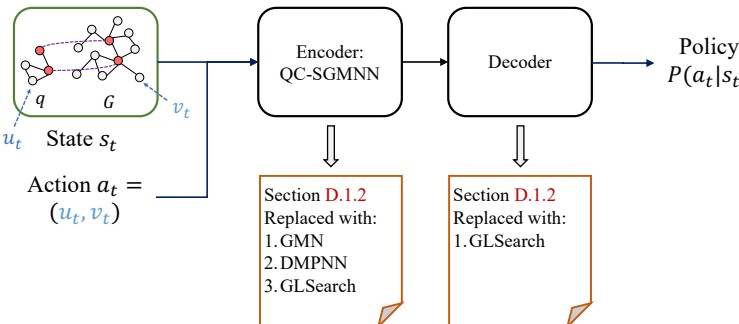

Figure 5: Summary of ablation studies performed in Section D.1.2. For GMN and DMPNN, only the encoder design is changed to other models, and all the other components, including the decoder choice, the search method, the training data and scheme, etc. are exactly the same as our proposed NSUBS.

the encoder training is shown to be important, we then perform a series of ablation study experiments on the encoder.

4. NSUBS-2-encoder-layers uses 2 encoder layers instead of 8. NSUBS-$D = 4$ decreases the embedding dimension from 8 to 4. NSUBS replaces the GRAPHSAGE model with GIN. These 3 models perform worse than NSUBS, among which NSUBS-$D = 4$ performs the worst, indicating that a dimension that is too small cannot capture enough information for the HPRD dataset.

5. NSUBS-no-Q-matching removes the concatenation of $h_{q,\text{intra}}^{(k+1)}$ in the computation of $h_{v,q \to G}^{(k+1)}$ in Equation 4. The worse performance shows the contribution of making the attention from $q$ to $G$ conditioned on the whole-graph representation of $q$.

6. NSUBS-no-LDP-encode removes the Local Degree Profile encoding mentioned in Section A.4. The worse performance shows the importance of encoding the nodes initially with degree information.

### D.1.2 ENCODER-DECODER REPLACEMENT

In addition, we perform the ablation study by replacing our QC-SGMNN encoder with the Graph Matching Neural Network (GMN) (Li et al., 2019) encoder. GMN introduces the idea of inter-graph message passing, but our novelties make it feasible, in terms of efficiency and effectiveness, for Subgraph Matching.

We summarize three key differences between GMN and QC-SGMNN:

- GMN produces a soft node-node matching matrix between two graphs instead of hard node selections indicating which nodes should be matched between $q$ and $G$. We therefore only utilize GMN's encoder layer and everything else being the same as our model, including the decoder, the training scheme, the search framework, etc. In other words, we adapt GMN for Subgraph Matching denoted as GMN-encoder in Table 3.

- To improve GMN's efficiency, QC-SGMNN (1) prunes the intergraph interaction space using the local candidate set and (2) decouples the intra- and inter- message passing for caching. Section D.2 provides more details.

- To improve GMN's effectiveness, QC-SGMNN (1) prunes noisy inter-graph interactions with the local candidate set and (2) proposes carefully designed graph-to-node propagation.

The worse performance of GMN indicates the necessity of the proposed QC-SGMNN designed for Subgraph Matching.

Although DMPNN outputs an approximate count between $q$ and $G$, we adapt DMPNN's encoder in our framework, and the worse performance of DMPNN-encoder again validates the usefulness of QC-SGMNN.

Besides what is mentioned in Section C.3, there are some additional notes regarding how we adapt GLSEARCH. To adapt GLSEARCH for subgraph matching, we first convert GLSEARCH's value network into a policy network by pipelining (1) an action execution engine, which executes each action, $a_t$, in the action space for an input state, $s_t$, to obtain a set of next states $\{s_{t+1}|\forall a_t \in \mathcal{A}_{u_t}\}$, (2) the GLSEARCH value network, which converts each next state, $s_{t+1}$, into a single policy network score, $p(a_t|s_t)$.

Generalizing the equivalence class idea from GLSEARCH to subgraph matching, we use the local candidate sets in place of bidomains within the adapted GLSEARCH. Notice, since GLSEARCH does not create node embeddings for action, $a_t$, we remove the max margin component of lookahead loss for these experiments. We use the same graph neural network encoder type as QC-SGMNN, but unlike NSUBS, which only runs the graph neural network once, GLSEARCH runs the graph neural network for each action in the action space. Hence, to prevent out of memory errors, GLSEARCH only uses 3 layers instead of 8. We use the same hyperparameters for interact and pooling model architecture, dimensions, and strides as described in the GLSEARCH paper. We call this model GLSEARCH-adapted.

The worse performance of GLSEARCH-adapted verifies NSUBS's choice of policy network, model design, and loss greatly improves performance.

## D.2 REDUCTION OF CANDIDATE SPACE

We plot the change of global candidate space (CS) size and the local candidate space (LCS) size across time in Figure 6. Specifically, at each search iteration, a node $u_t$ from the query graph is chosen by $\phi$, and the local candidate space is computed for $u_t$. Therefore, we can record the number of candidates for $u_t$ in the global candidate space and in the local candidate space, and plot the change of CS and LCS sizes across search iterations. As shown in Figure 6, there is a large reduction of candidate nodes from CS to LCS, suggesting the importance of computing a smaller local candidate space based on the current matching and the selected $u_t$. Without any LCS computation technique, the action space would be the global CS, which may contain up to $100,000$ nodes. Another important observation is that, the number of local candidate nodes varies dramatically across iterations. For example, in the initial iteration, since no nodes are matched, LCS is of the same size as CS, but as soon as one node pair is matched, the LCS computation can leverage the matched nodes by narrowing down the action space from all the nodes in CS to the first-order neighbors of the matched nodes. Existing LCS computation has several other techniques to reduce the candidate space size other than simply using the first-

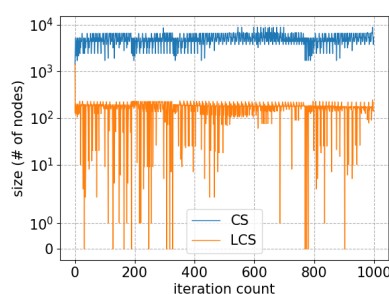

Figure 6: Demonstration of the sizes of global candidate space (CS) and local candidate space (LCS) across search iterations, by running NSUBS on a randomly sampled graph pair from HPRD, and collecting the size of CS and LCS at each search iteration. The y-axis is in the log scale. We stop recording the CS and LCS sizes after 1000 search iterations.

order neighbors of the matched subgraphs, and as a result, there can be less than 10 nodes in $\mathcal{A}_{u_t}$ in some iterations. However, when the search backtracks to an earlier state, there can be more candidate nodes in LCS, resulting in the varying LCS sizes across search.

The reduction of candidate space size from CS to LCS confirms the proposed usage of LCS to compute message passing between $q$ and $G$, i.e. the computation of $\tilde{M}_t$ and $\tilde{M}_t^{-1}$ used in Equation 4. The GMN model in contrast does not consider the state-dependent LCS, and uses all the quadratic node-node pairs in the cross-graph message passing mechanism, as explained in the ablation study (Section D.1).

## D.3 PERFORMANCE ACROSS ITERATION

We show the number of solved pairs across search iterations in Figure 7, similar to Figure 2 except for the x-axis which is replaced with search iterations. It is noteworthy that all the methods are

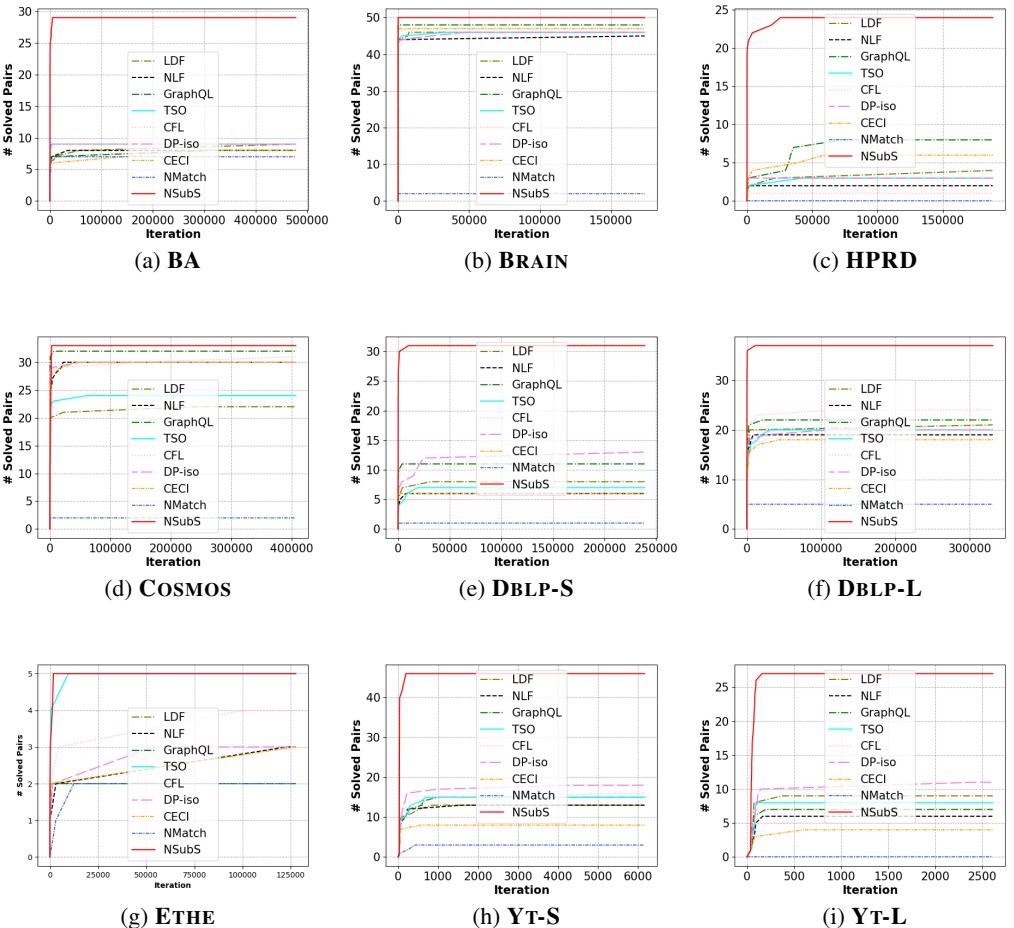

Figure 7: The growth of the number of solved pairs across search iterations.

implemented under the same search framework, and therefore the search iteration counting and measurement are consistent across the methods.

### D.4 PERFORMANCE W.R.T. QUERY SIZES

We show the performance of each method under different query graph sizes for BA and HPRD. As shown in Table 4 and 5, the difficulty of solving Subgraph Matching increases as the query graph sizes increase. When the query graph only contains 8 nodes, all the methods can solve all the graph pairs under the 5 minutes time budget. However, as $|V_q|$ increases to 128 nodes, all the methods fail to solve any graph pair, suggesting the room for improvement. In the main text, we report the results for $|V_q| = 64$, which is of moderate difficulty. Under other graph sizes, the proposed NSUBS performs either the best or very close to the best method highlighted in bold.

## E RESULT VISUALIZATION

### E.1 ILLUSTRATION OF SEARCH TRAJECTORY

We first show an animation of the actions selected by NSUBS across search in Figures 9 10, and 11. For each plot, we (1) find the search branch with the largest found subgraph across the search tree, then (2) plot the actions executed in that search branch. For each action, we plot the resulting state, with the query graph on the left and target graph on the right. For clarity, we do not plot the whole

Table 4: Solved percentage ("Sol%") and number of solutions ("#Sol") found by each method after 5 minutes averaged by the number of the graph pairs in the BA dataset by varying $|V_q|$ ("BA-$|V_q|$"). For clarity and compactness, the number of solutions result has been divided by 1000, i.e. each number is in the unit of $10^3$. For BA-128, since all the methods fail to solve any pair, we report the average ratio of the largest match solution with respect to the query graph size denoted as "Ratio".

| Method | BA-8 Sol% | #Sol | BA-16 Sol% | #Sol | BA-32 Sol% | #Sol | BA-64 Sol% | #Sol | BA-128 Sol% | Ratio |
|---|---|---|---|---|---|---|---|---|---|---|
| LDF | 1.00 | 14.11 | 0.86 | 10.70 | 0.74 | 7.21 | 0.18 | 1.21 | 0.00 | 0.19 |
| NLF | 1.00 | 14.30 | 0.92 | 11.18 | 0.72 | 6.60 | 0.16 | 1.29 | 0.00 | 0.23 |
| GRAPHQL | 1.00 | 14.21 | 0.94 | **12.03** | 0.78 | 7.64 | 0.16 | 1.42 | 0.00 | 0.16 |
| TSO | 1.00 | 14.15 | 0.64 | 11.91 | 0.74 | 6.95 | 0.18 | 1.41 | 0.00 | 0.18 |
| CFL | 1.00 | 14.30 | 0.94 | 11.82 | 0.76 | 7.13 | 0.18 | 1.56 | 0.00 | 0.16 |
| DP-ISO | 1.00 | 14.27 | 0.94 | 11.71 | 0.82 | 7.35 | 0.18 | 1.31 | 0.00 | 0.17 |
| CECI | 1.00 | 14.00 | 0.92 | 11.57 | 0.72 | 6.91 | 0.16 | 1.21 | 0.00 | 0.17 |
| NMATCH | 1.00 | **14.55** | 0.88 | 11.35 | 0.66 | 6.18 | 0.14 | 1.02 | 0.00 | 0.14 |
| NSUBS | 1.00 | 13.06 | **1.00** | 12.01 | **0.98** | **8.27** | **0.62** | **2.37** | 0.00 | **0.35** |

Table 5: Performance of each method by varying $|V_q|$ ("HPRD-$|V_q|$") for HPRD.

| Method | HPRD-8 Sol% | #Sol | HPRD-16 Sol% | #Sol | HPRD-32 Sol% | #Sol | HPRD-64 Sol% | #Sol | HPRD-128 Sol% | Ratio |
|---|---|---|---|---|---|---|---|---|---|---|
| LDF | 1.00 | 14.50 | 0.92 | 11.92 | 0.64 | 6.48 | 0.08 | 0.69 | 0.00 | 0.13 |
| NLF | 1.00 | 14.74 | 0.96 | 12.28 | 0.64 | 6.51 | 0.04 | 0.37 | 0.00 | 0.13 |
| GRAPHQL | 1.00 | 14.33 | **1.00** | 13.07 | **0.90** | **9.34** | 0.16 | 1.24 | 0.00 | 0.13 |
| TSO | 1.00 | 14.47 | 0.98 | 12.64 | 0.66 | 6.81 | 0.06 | 0.50 | 0.00 | 0.11 |
| CFL | 1.00 | 14.44 | 0.94 | 12.12 | 0.58 | 5.63 | 0.02 | 0.18 | 0.00 | 0.11 |
| DP-ISO | 1.00 | 14.28 | 0.96 | 12.44 | 0.68 | 6.85 | 0.06 | 0.54 | 0.00 | 0.12 |
| CECI | 1.00 | 14.44 | **1.00** | **13.13** | 0.64 | 6.28 | 0.12 | 1.05 | 0.00 | 0.12 |
| NMATCH | 1.00 | **16.06** | 0.82 | 10.98 | 0.32 | 3.03 | 0.00 | 0.00 | 0.00 | 0.12 |
| NSUBS | 1.00 | 13.71 | **1.00** | 12.68 | **0.90** | 9.10 | **0.50** | **4.17** | 0.00 | **0.19** |

target graph, instead a fixed percentage of one-hop and two-hop neighbors of that target graph. We make the following deductions based on the animations. The colors of nodes shows the mapping.

First, NSUBS tends to pick high degree nodes in the target graph as shown in Figure 8. This makes sense as every edge in the query must match some edge in the target. High degree nodes have more edges to satisfy this constraint. Existing works also find high correlation between static geometric representations and node degree (Lou et al., 2020), indicating the geometric component of lookahead loss is working. A natural question is then whether a simple degree based policy would work equally well or even better than the proposed model. To confirm the necessity of learning-based policy, we find that setting the raw output of policy to the degree of $v_t \in \mathcal{A}_{u_t}$, $P_{\text{raw}}((u_t, v_t)|s_t) = \text{Deg}(v_t)$, solves 36% of the pairs in HPRD, which is worse than NSUBS yet better than many baselines as in Table 3. Since existing baseline methods place focus on improving node selection in $q$, i.e. the ordering method $\phi$ for $V_q$, the selection of nodes in $G$ is random (Section 2.3). Our work is the first to improve selection from $V_G$, suggesting a new direction for tackling Subgraph Matching.

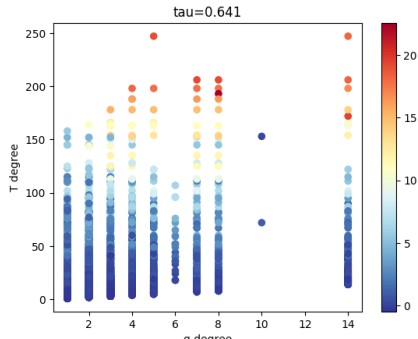

Figure 8: Correlation between node degree and the raw policy output before normalization learned by NSUBS. The Kendall's $\tau$ coefficient (Kendall, 1938) between the two is shown on the top.

NSUBS tends to map hub nodes in the query to hub nodes in the target. This suggests QC-SGMNN effectively captures structural properties of the graph pair, as the model can look beyond 1 hop neighborhoods to determine whether the node should be in a hub or just a dangling high degree node.

NSUBS tends to map dangling degree one nodes in the query to low degree nodes in the target. This makes sense in the context of search, as matching a low degree query node to high degree target node means said higher degree target node cannot be matched later in search on query nodes with more constraints. This indicates the RL component of lookahead loss is working.

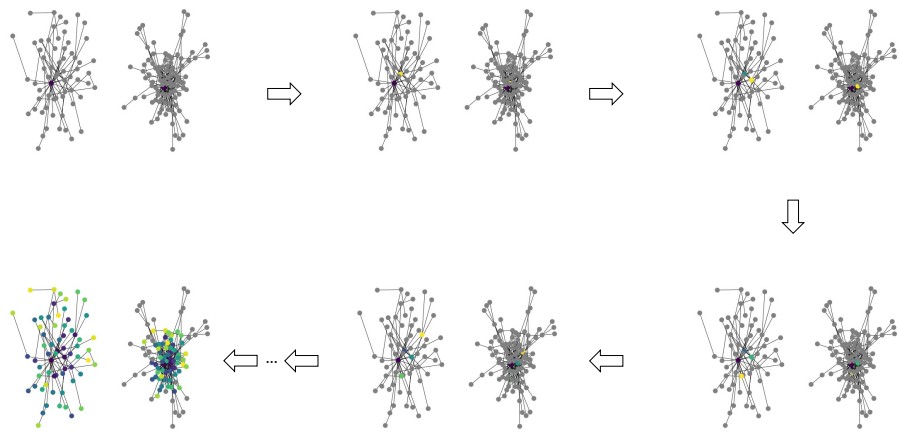

Figure 9: Illustration of the search trajectory on a graph pair pair of BA.

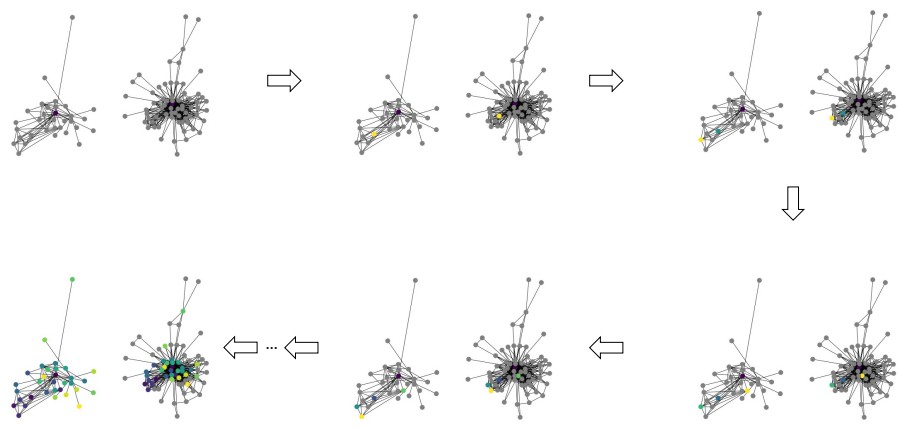

Figure 10: Illustration of the search trajectory on a graph pair pair of COSMOS.

### E.2 VISUALIZATION OF SOLVED PAIRS

We show five instances of solved graph pairs by NSUBS in Figures 12, 13, 14, 15, 16, 17, 18 and 19. We first plot (1) the query graph on its own, then plot (2) the query graph matched to the target graph, and finally plot (3) the target graph.

In other words, the first and second plots correspond to the same query graph but their node positions/layouts are different due to different layouts. In the second plot we fix the positions of nodes in $q$ to match the positions of their matched nodes in $G$, i.e. the nodes in the second and the third plots have the same relative node positions for visualizing the node-node mapping. The colors of nodes are

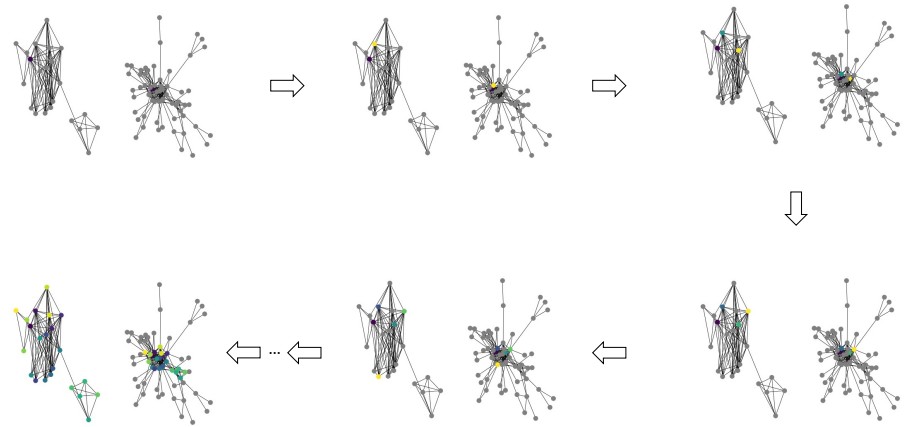

Figure 11: Illustration of the search trajectory on a graph pair pair of DBLP.

for the purpose of visualizing the mapping. For clarity, we only show a subgraph of $G$ instead of the entire $G$ as it contains too many nodes and edges to show. Specifically, we include the matched $q$ in $G$ and grow the matched subgraph by including the first-order neighbors of the matched nodes.

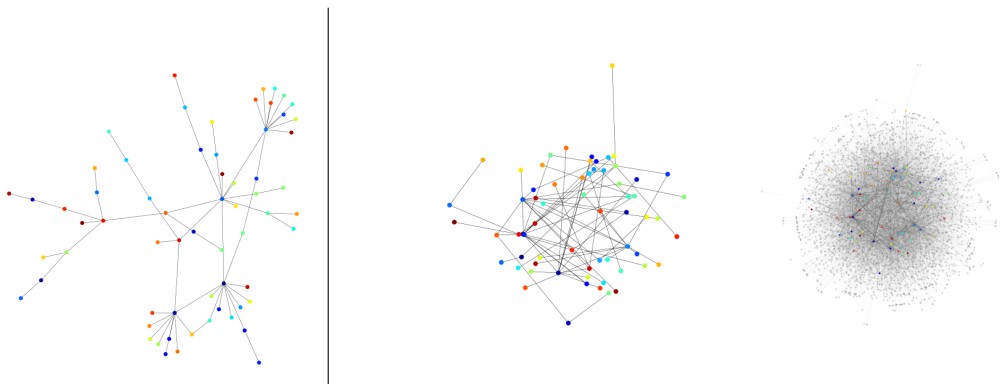

Figure 12: Visualization of a solved pair on BA.

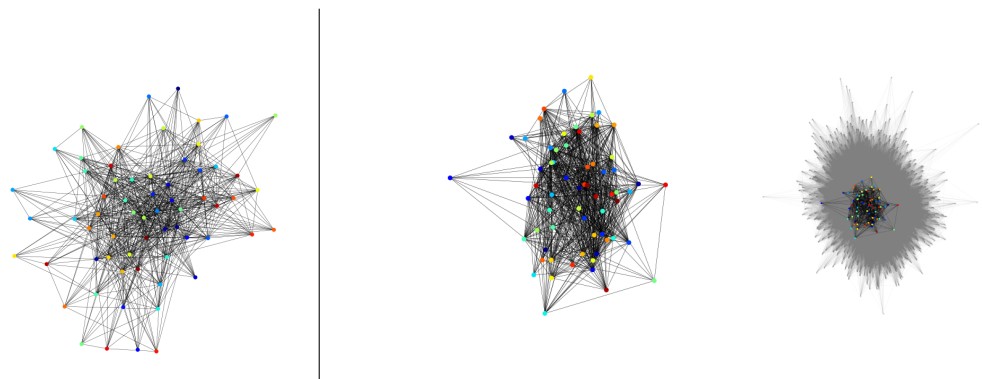

Figure 13: Visualization of a solved pair on BRAIN.

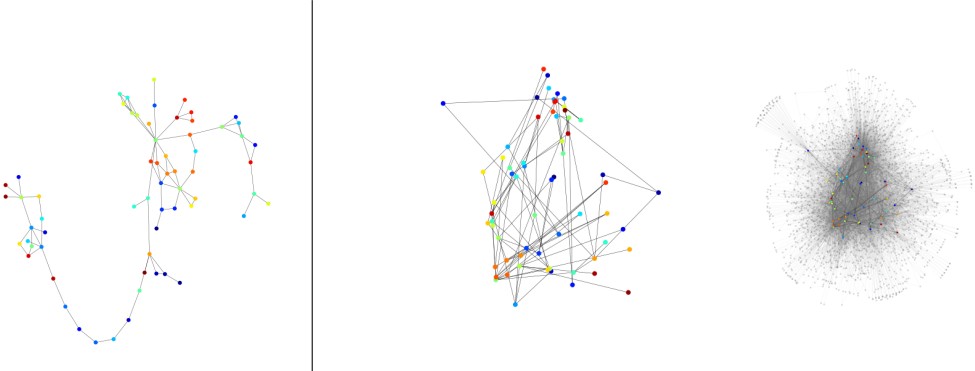

Figure 14: Visualization of a solved pair on HPRD.

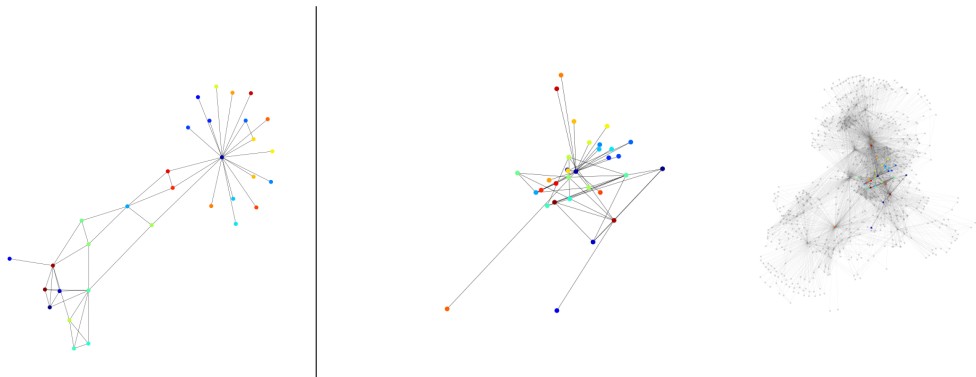

Figure 15: Visualization of a solved pair on COSMOS.

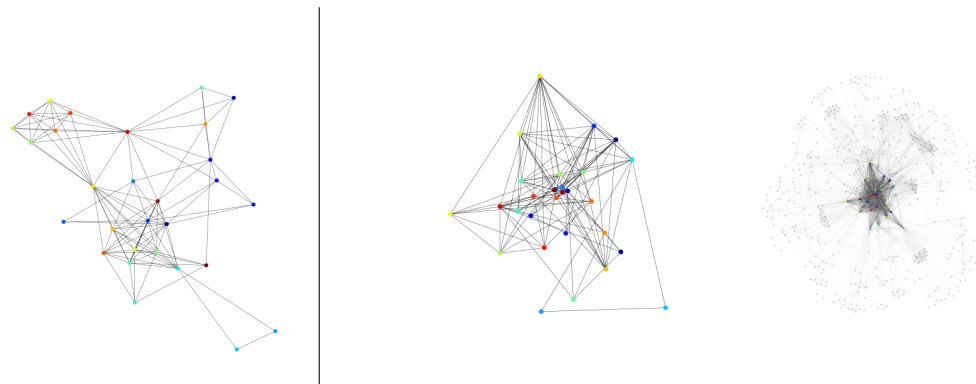

Figure 16: Visualization of a solved pair on DBLP.

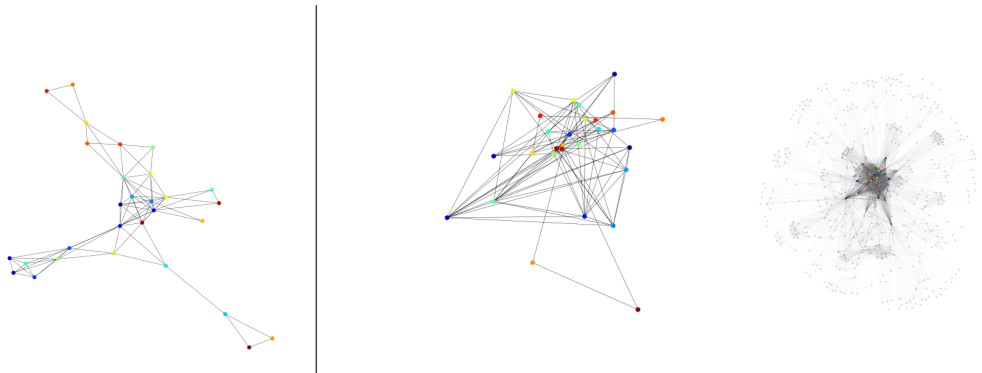

Figure 17: Visualization of a solved pair on DBLP.

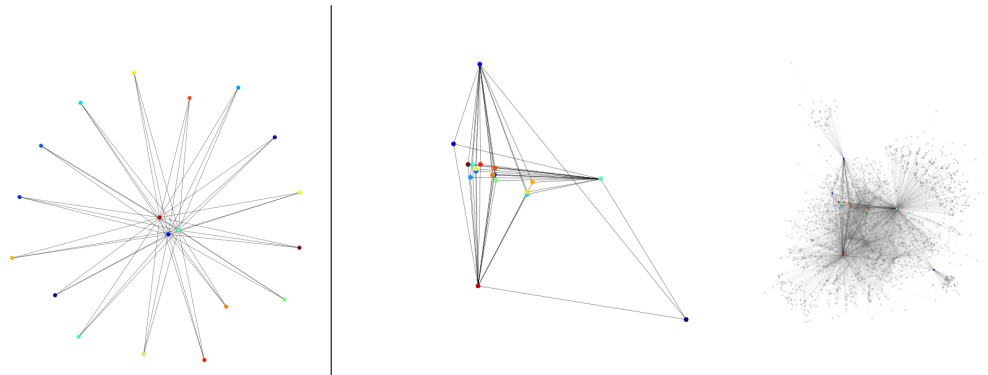

Figure 18: Visualization of a solved pair on ETHE.

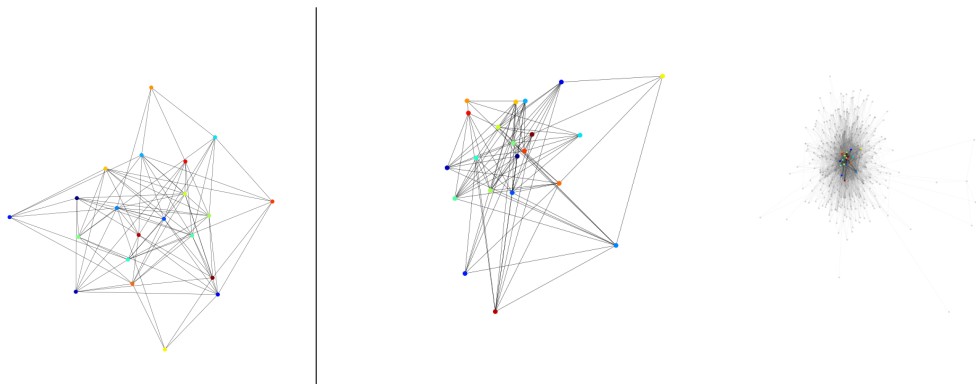

Figure 19: Visualization of a solved pair on ETHE.

