# OpenReview forum: "Detecting Small Query Graphs in A Large Graph via Neural Subgraph Search"
_ICLR.cc/2023/Conference — Submitted to ICLR 2023_

### Official Review · Reviewer_HPUs · 2022-10-27

**Confidence:** 3
**Correctness:** 3
**Technical Novelty And Significance:** 2
**Empirical Novelty And Significance:** Not applicable
**Recommendation:** 6

**Clarity, Quality, Novelty And Reproducibility:**

The paper is well-written and easy to follow. The quality is good except I would love to see a few more experiments. Novelty is fair.

**Strength And Weaknesses:**

The paper is well-executed and easy to follow. All parts are clearly presented and convincing. Experiences are thoroughly done.
The key contribution also involves the dynamic message passing + a look-ahead loss to guide the RL to find the optimal order of search.

1. It is great that the authors provided time complexity and empirically validated it via experiments. But It is unclear the training complexity. Especially with the look-ahead loss, do we have to add up many future steps to compute the loss at every step?
2. It would be helpful in the experiments the authors could provide a total time for training the learnable models, including this model. For models that often get stuck in a local minimum and often do not improve further after the first 20-60 seconds (fig 2), would it be fair to restart them with some different random seeds and repeat them until 5min?
3. I was a bit confused: does NSUBS also follow a general DFS + backtracking approach? If so, wouldn't it suffer from the local minimum problem as well?
4. The authors claimed NSUBS works better at large query graphs. But the results section doesn't provide further performance stratifications on queries w. different sizes. It would strengthen the paper if the readers can see some breakdown numbers to better understand this claim.
5. Related to 4, it would be helpful if the authors could have some effort into the model interpretation. For example, show an actual search trajectory and show how the NSUBS modifies the local candidate list to speed up and optimize the search.

**Summary Of The Paper:**

The authors presented a neural search method NSUBS to solve the subgraph search problem. The key idea is to pass information between the query and a local candidates pool at every search step to optimize the search.

**Summary Of The Review:**

I am willing to increase my score if the authors could address my questions and provides some more fair comparison and insights/limitations of the model.

---

> ### Author Response · Authors · 2022-11-15
> **Thank you for your reviews!**
>
> Dear Reviewer,
>
> The lookahead loss does not introduce extra complexity. With standard RL loss, one would complete the search process, push state-action pairs to the replay buffer, then execute the loss using the replay buffer. Lookahead loss grabs positive negative pairs for each state-action pair by following a pointer to the future search state that finds the largest subgraph matching. No additional search iterations are required as said pointers can be efficiently updated during search and pushing to the replay buffer happens after search.
>
> We have shown the performance of all the methods with the query graph sizes varying from 8 to 128 for the BA and the HPRD datasets. The results are added as Tables 4 and 5 in the revisioned paper. In short, we find all the methods perform worse when query graph sizes increase, yet our proposed NSubG still outperforms baselines even when the query graph is as large as 128 nodes.

---

> > ### Author Response · Authors · 2022-11-17
> > **Performance under different query graph sizes**
> >
> > We evaluate NSubS and baselines on different query graph sizes. As expected, NSubS finds more solutions on larger/more difficult graph pairs. On small and easy pairs, all methods can find the solution since all can iterate through large portions of the search space within 5 mins. On large and difficult pairs, the search space is too large, hence NSubS, which picks promising actions first, becomes more effective. Our experiments show NSubS becomes efficient when query graphs are larger than 8 nodes. Note, while all methods fail to find exact matchings after 128 node queries, NSubS still finds significantly larger partial subgraph matchings than baselines. we also find the same trends on the HPRD dataset and provide both BA and HPRD results in Section D.3 of the Supplementary Material.

---

> > > ### Author Response · Authors · 2022-11-19
> > > **Actual search trajectory plotted**
> > >
> > > We have plotted three actual search trajectories as Figures 9, 10, and 11 and propose some model interpretation in Section E.1 of the revisioned paper. Thank you very much for your reviews.

---

### Official Review · Reviewer_AyD2 · 2022-10-28

**Confidence:** 4
**Correctness:** 3
**Technical Novelty And Significance:** 3
**Empirical Novelty And Significance:** 2
**Recommendation:** 6

**Clarity, Quality, Novelty And Reproducibility:**

The paper presents some novel ideas. Certain parts need to be better explained. Practical impact (motivation/applications/results) may not be as significant as one would have hoped.

**Details Of Ethics Concerns:**

nil

**Strength And Weaknesses:**

Strengths:
1. The idea of inter-graph message passing for matching is novel to me. Existing GNN architectures only focus on the intra-graph counterpart.

2. The use of reinforcement learning can reduce or eliminate labeling effort.



Weaknesses:
1. There is a large body of literature on the related subgraph isomorphism counting problem. In many applications, the profile of subgraph counts can sufficiently characterize a graph and it is not necessary to find all or most instances of the query.  Discussing some case studies and example applications could be useful to motivate why it is necessary beyond counting.

2. The proposed model has high initial overhead. The methods only show significant advantage after a few mins. However, in many cases it might be important to get a few matching instances fast rather than getting many instances later on.

3. Some parts are not well explained in detail.
- 3.1: "We define our reward as R(+) = 1 if the subgraph is fully matched and R(−) = 0 otherwise." Do you actually mean the "current" subgraph is fully matched?
- 3.1: "Therefore, after the search, we collect the positive training signals at each state as all the node-node pairs (u, v) that lead to a solution" I'm not clearly how this is exactly done. A step-by-step explanation might be useful.
- 3.2.1: How are the mapping M, M' are constructed? Again, a step-by-step instruction might be clearer.

Minor issues:
The ablation study is important and should be moved to the main paper.

**Summary Of The Paper:**

The paper proposes a learning based method for subgraph matching, using a graph neural network-based encoder-decoder architecture, and a reinforcement learning strategy. The GNN consists of two main modules: intra-graph message passing for modeling graph structures, and inter-graph message passing for query-graph matching. Furthermore, a look-ahead loss is formulated to supplement training signals.

**Summary Of The Review:**

Overall I think the weaknesses slightly overweigh the strengths.

---

Update: I'm largely satisfied with the responses, except for the initial overhead. The cost of executing NNs is part of the cost, which cannot be simply discounted. Overall, i will upgrade my score.

---

> ### Author Response · Authors · 2022-11-12
> **Thank you for your review!**
>
> Thank you for your insightful review!
>
> (1) In circuit analysis for example, finding the match locations can be more useful than knowing the count of isomorphic subgraphs, because circuit designers can directly modify and analyze the schematic at said matched locations. In addition, in detecting anomalous subgraphs in a financial transaction network, e.g. the Ethereum dataset used in our paper, returning the exact nodes that correspond to potentially malicious accounts is much more important and useful than simply knowing if or how many such suspicious subgraphs exist within the very large Ethereum transaction network.
>
> (2) The high initial overhead is purely due to the extra runtime incurred by executing the neural network. In terms of search iterations, NSubS does not incur any extra overhead vs. baselines. The matching instances that the baselines solve faster than NSubS are easier queries, whose search space is heavily pruned by local candidate sets. The matching instances that NSubS solve faster than baselines are harder queries, where the search space is too large for simple enumeration or weak policies to succeed. Looking at the runtime plots (Figure 2), the intersection between NSubS’ line and the baseline’s line denotes the boundary between these “easy” and “hard” cases. While NSubS is able to solve easy cases given reasonable additional running time (few mins), the baselines require exponentially more time to solve hard cases, due to the NP-hard nature of subgraph matching.
>
> This can further be seen in Tables 4 and 5, which show baselines begin failing when the query graphs become larger than 8 nodes. These suggest that the learned policy is indeed “smarter” than baselines, and the performance gain of the learned policy over the best baseline only becomes larger when the query becomes more difficult, as shown below.
>
> ```
>             BA-8         BA-16       BA-32        BA-64       BA-128
>          Sol% #Sol    Sol% #Sol    Sol% #Sol    Sol% #Sol    Sol% Ratio
> Gain      0   -1.49   0.06  -0.02  0.16  0.63   0.44  0.81   0     0.12
> ```
>
> Last but not least, the initial performance drop compared to the baseline due to the query being too small and quick to solve can be alleviated using existing techniques to improve the inference speed of GNNs, such as using C++ to implement the neural network, using knowledge distillation to obtain a faster model (Zhang, Shichang, et al. "Graph-less neural networks: Teaching old mlps new tricks via distillation." arXiv preprint arXiv:2110.08727 (2021).), etc.
>
> (3) To clarify the details you mentioned:
>
> * 3.1 If the “current” subgraph eventually becomes fully matched by search, then R(+) = 1. Otherwise, R(-)=0.
>
> * 3.2 The replay buffer is populated after search runs for a fixed timeout. Each search state in the search tree is sent to the replay buffer. For a given search state, we find all subgraph matches found that originated from that search state. All actions, (u,v), that exist in those eventual subgraph matches are positive training signals.
>
> * 3.3 M_t is initialized as an empty set {}. Each time an action, a_t=(u,v), is executed, it is appended to M_t to form M_{t+1}. M_t’ is initialized as the global candidate set (line 3 of Algorithm 1) and is updated each time an action is executed (line 8 of Algorithm 2) to ensure isomorphism constraints are satisfied, as detailed in https://dl.acm.org/doi/10.1145/3318464.3380581.
>
> We will update the final paper to make these details more clear.

---

> > ### Author Response · Authors · 2022-11-19
> > **Further details on (1), (2), and (3)**
> >
> > We added a step-by-step explanation on how the training signals are collected in Section A.1 of the Supplementary Material.
> >
> > We added a step-by-step animation for M_t and M_t' in Figure 9 and 10 of the Supplementary Material. M_t denotes the current subgraph matching at each search state (i.e. selected colored nodes). While not explicitly shown, M_t' denotes unmatched nodes that can be matched without breaking the isomorphism constraints (i.e. all gray nodes in the query, and some gray nodes in the target, because over time unmatched target nodes may no longer be match-able to any query nodes).

---

### Official Review · Reviewer_181W · 2022-11-02

**Confidence:** 3
**Clarity, Quality, Novelty And Reproducibility:** 1. The authors claim that heuristics-…
**Correctness:** 3
**Technical Novelty And Significance:** 3
**Empirical Novelty And Significance:** 3
**Recommendation:** 5

**Strength And Weaknesses:**

Strengths:
1. Detailed preliminary introduction to help the readers to understand the pipeline.
2. The problem is not easy and deserves more attention.
3. The proposed method is technically sound.

Weaknesses:
1. As far as I know, the related work for learning-based methods for subgraph matching is incomplete.
2. The paper contains many notations and is hard to follow. Some preliminary knowledge could be moved to the Appendix because the main focus of this paper is on neural design.
3. The methodology contribution is limited. Several works have adopted the idea of designing the encoder to encode two graphs and the design of the decoder is not novel.


**Summary Of The Paper:**

The paper studies the subgraph matching problem to perform a subgraph isomorphism check between a query graph and a large target graph. The authors attempt to solve the solution and reward sparsity issues by designing an encoder-decoder neural network and look-ahead loss function based on reinforcement learning. Experiments conducted on six datasets show the effectiveness.

**Summary Of The Review:**

The paper proposes a reinforcement learning framework to solve the subgraph matching problem and designs an encoder and a decoder. The paper is technically sound but missed several related works and is not easy to follow. Also from my perspective, the methodology contribution is limited.

---

> ### Author Response · Authors · 2022-11-12
> **Thank you for your review!**
>
> Thank you for your clear feedback! We summarize the relationship between our work and the works listed in the Related Works common question. Generally, the papers listed are either enhanced or orthogonal to our work. Please check the Related Works common question about this and let us know if you still have concerns!
>
> Our encoder design includes several high-impact novelties that extend beyond subgraph matching, including: (1) the idea of filtering noisy intergraph messages, (2) using graph-to-node propagation to overcome any information bottlenecks introduced by filtering, and (3) decoupling intra- and inter-graph message passing for caching. Please see our encoder novelty common question for details.
>
> In addition to encoder design, our loss function is also significant, unifying the many one-shot geometric similarity learning literature (https://arxiv.org/abs/2007.03092, https://arxiv.org/abs/2001.09621, https://ojs.aaai.org/index.php/AAAI/article/view/20784) and reinforcement learning literature (https://arxiv.org/abs/1704.01665, https://arxiv.org/abs/2002.03129). Please see our loss function novelty common question for details.
>
> Regarding the heuristic-based ordering to select nodes in q, we agree that the current framework adopts a heuristics for query vertex ordering, and learns a policy for node selection for G (the target graph). For large graphs, target-node ordering becomes the bottleneck instead of query node-ordering, as enumerating all target nodes for each query vertex becomes infeasible by the NP-Hard nature of subgraph matching.
>
> From the Weisfeiler-Lehman perspective, graph neural networks can approximate but not perfectly solve the graph isomorphism problem (https://arxiv.org/abs/1810.00826). In our case, we learn geometrically-informative representations to enhance the speed of graph matching search, but this does not mean NSubS guarantees every graph pair can be solved in polynomial iterations -- only that it can speed up the search process by generalizing representations learned during training to the test set. Experimentally, we find the candidate space to contain hundreds of candidate nodes, so a weak policy is unlikely to match the correct nodes, and we show on seven target graphs and nine datasets our proposed policy is able to outperform existing methods.
>
> Notice, the target graph can have millions of nodes. In this case, summing millions of node embeddings introduces too much noise (https://arxiv.org/abs/1810.00826). A better approach would be gradual graph coarsening such as (https://arxiv.org/abs/1806.08804, https://arxiv.org/abs/1904.08082, https://arxiv.org/abs/1905.05178, https://arxiv.org/abs/1907.00481), but QC-SGMNN already does gradual graph coarsening through inter-graph message propagation implictly. Notice, the pooled query graph nodes already contain information from all the relevant target graph nodes (target graph nodes that belong to a local candidate set), because each query graph node receives messages from all relevant target graph node embedding in the previous round of intergraph message passing. Target graph nodes not in the local candidate set can never be picked in future search iterations, so the exclusion of these nodes is a favorable invariance captured by QC-SGMNN. Hence, QC-SGMNN completely preserves the target graph information.

---

> > ### Author Response · Authors · 2022-11-17
> > **We provide experimental results for GMN, DMPNN, and GLSearch**
> >
> > Sicne GMN and DMPNN do not provide discrete subgraph matching, we adapt them under the NSubS framework to test the novelty of our QC-SGMNN encoder. Although GLSearch’s maximum common subgraph (MCS) framework can be applied for subgraph matching in theory, search algorithms designed specifically for subgraph matching work much better than those designed for MCS in practice (https://link.springer.com/chapter/10.1007/978-3-030-19212-9_2), thus we also adapt GLSearch under the NSubS framework, to test our QC-SGMNN encoder and Lookahead loss function.
> >
> > To show the effectiveness of (1) intergraph message filtering, (2) query conditioning, and (3) caching, we replace the QC-SGMNN encoder with the GMN encoder. As seen below, QC-SGMNN outperforms GMN by filtering out noisy intergraph messages, introducing state dependency by filtering based on a state’s local candidate set, and improving computational efficiency.
> >
> > To show the effectiveness of (1) intergraph message passing and (2) query conditioning, we replace the QC-SGMNN encoder with the DMPNN encoder. As seen below, QC-SGMNN outperforms DMPNN by interacting the graph pair and introducing state dependency by filtering intergraph messages based on a state’s local candidate set.
> >
> > To show NSubS’ novelty, we remove lookahead loss and replace QC-SGMNN with GLSearch’s value network. We also replace GLSearch’s bidomain partitioning with its subgraph matching equivalent, local candidate set. As seen below, both novelties greatly improve the performance of NSubS over GLSearch.
> >
> > ```
> >                                    HPRD
> >                                 Sol%   #Sol
> > NSubS                           0.48   4.17
> > GMN-encoder                     0.30   2.89
> > DMPNN-encoder                   0.32   2.25
> > NSUBS-no-look-ahead             0.42   3.68
> > GLSearch-encoder                0.22   1.88
> > ```
> >
> > Please note that we added these results to Section D.1 of the Supplementary.

---

### Official Review · Reviewer_vmC2 · 2022-11-04

**Confidence:** 3
**Correctness:** 3
**Technical Novelty And Significance:** 1
**Empirical Novelty And Significance:** 1
**Recommendation:** 5

**Clarity, Quality, Novelty And Reproducibility:**

The paper is written clearly. However, the quality of the experiments are poor.
The code is not given with the paper. Also, novelty of the paper is limited as it follows a very close approach with https://arxiv.org/abs/2002.03129.

**Strength And Weaknesses:**

Strengths:
+ Important problem
+ Intuitive approach

Weakness:
- Lack of well known baselines
- Lack of novelty

**Summary Of The Paper:**

The paper provides a reinforcement learning method for subgraph matching problem. Given a query graph, they attempt to find the subgraph from a large graph that matches a query graph. Then the paper compare the underlying method against several baselines on real world datasets.

**Summary Of The Review:**

I have two key concerns with the paper.
First, lack of powerful baselines. The paper acknowledges that most baselines are supervised learning methods. Thus, they adapted it into the RL method, which we appreciate. In this context, there are two powerful baselines which are never used:
(1) https://arxiv.org/abs/1904.12787 and
(2) https://ojs.aaai.org/index.php/AAAI/article/view/20784

Second, I really did not find any significant difference (of course there is some difference but they are not significant) from the approach adapted in https://arxiv.org/abs/2002.03129.

---

> ### Author Response · Authors · 2022-11-12
> **Thank you for your review!**
>
> Thank you for your valuable input! Our problem statement tackles subgraph matching on any query and target graphs including very large ones, which requires extracted subgraphs to satisfy several constraints, including (1) matched subgraphs are isomorphic to each other, (2) matched subgraphs are connected, (3) matched nodes have the same node labels.
>
> Regarding (1), https://arxiv.org/abs/1904.12787 (GMN), our work is inspired by the intergraph message passing idea introduced in (1); however, (1) is more comparable to our QC-SGMNN graph neural network than our NSubS framework, since it only provides a soft matching matrix, where we require a discrete query to target graph matching. Our QC-SGMNN extends GMN by (A) using a local candidate space to filter out noise from the O(|Gq|*|Gt|) intergraph messages and condition GMN on the search state, (B) introduce whole graph embedding messages to overcome any information bottlenecks introduced by filtering, and (C) decoupling intra- and inter- message passing to speed up (1) with caching. Please see our Novelty common question for a thorough comparison.
>
> Regarding (2), https://ojs.aaai.org/index.php/AAAI/article/view/20784 (IsoNet), we appreciate their contribution towards using edge-centric geometric representations; however, (2) is not directly comparable because it only provides a soft matching matrix, where we require a discrete query to target graph matching. Because we must find the query subgraph matching, not just the ranking, we leverage the state-of-the-art subgraph matching search algorithm, which is node centric. To the best of our knowledge, there are not any competitive edge centric search algorithms. Since (2) operates on edge-edge ranking scores instead of node-node policy scores, we propose lookahead loss, which extends a node-centric version of (2)’s novel loss function under the RL framework to better solve the subgraph search problem. Please see our Novelty and Related Works common question for a thorough comparison.
>
> Although the Hungarian algorithm can be applied to the matching matrix obtained by IsoNet, such an algorithm cannot guarantee the isomorphism constraint required by the Subgraph Matching task, i.e. the resulting one-to-one mapping between the query and target graphs cannot ensure the query and the matched subgraph are isomorphism. This is because the Hungarian algorithm aims to minimize the total cost while ensuring the one-to-one correspondence, and it is not straightforward to ensure an additional constraint that the node-to-node mapping results in subgraph isomorphism (since otherwise, the NP-hard problem of Subgraph Matching is solved by the polynomial-complexity Hungarian algorithm). In other words, we tackle the problem of Subgraph Matching by using a search algorithm to ensure the solution must obey the subgraph isomorphism constraint, while the IsoNet work produces a global node/egde alignment matrix which is only an intermediate result. Our "Experimental Results Regarding Isonet" response experimentally verifies this.
>
> Regarding https://arxiv.org/abs/2002.03129, GLSearch is designed for maximum common subgraph (MCS) rather than subgraph matching. Due to the task being different, we have design choices different from GLSearch search framework, model, and loss design. Concretely, our method exploits the asymmetric nature of subgraph matching vs MCS by (A) utilizing a more powerful search algorithm based on local candidate set (LCS) pruning, (B) proposing QC-SGMNN to improve the encoder by (B.1) filtering intergraph messages with LCS, (B.2) intergraph graph embedding propagation, and (B.3) decoupling the matching/propagation modules, and (C) proposing lookahead loss to improve RL loss by introducing geometric priors. More generally, NSubS uses a policy network which is more efficient than GLSearch’s value network. Please see our Novelty and Related Works common question for a thorough comparison.
>
> We have released our codebase to the supplementary material.

---

> > ### Author Response · Authors · 2022-11-19
> > **Major differences between GLSearch for MCS detection and NSubG for Subgraph Matching**
> >
> > NSubS is significantly different than GLSearch. While both methods use RL to solve a NP-Hard problem, the novelties proposed in NSubS more than double the performance of GLSearch as shown in Section D.1.2 of the Supplementary Details. We compare the two and list the major differences below. Please also refer to Section C.3 for details.
> >
> > ```
> > Method               |         GLSearch         |        NSubS
> > ---------------------------------------------------------------------------
> > RL Formulation       |       Value Network      |    Policy Network
> >                      |  (Factoring out action   | Efficient computation
> >                      |  to look at s_{t+1})     | of policy for action node
> >                      |                          | pair leveraging small local
> >                      |                          |   candidate space
> >                      |                          |
> > ---------------------------------------------------------------------------
> > Loss function        |         DQN loss         |       Lookahead loss
> > ---------------------------------------------------------------------------
> > Intra-graph message  |            Yes           |            Yes
> > passing in encoder?  |                          |
> > ---------------------------------------------------------------------------
> > Inter-graph message  |            No            |            Yes
> > passing in encoder?  |                          |
> > ---------------------------------------------------------------------------
> > Way to interact two  | Interact pooled bidomain |    Inter-graph message
> >       graphs         | embeddings after encoder |     message passing in
> >                      |                          |     each encoder layer
> > ---------------------------------------------------------------------------
> > Granularity of       |     Bidomain-level       |        Node-level
> > intergraph           |                          | (local candidate set
> > Interaction          |                          |     filtering) and
> >                      |                          |       graph-level
> > ---------------------------------------------------------------------------
> > Decoupling of        |     Not Applicable       |        Yes (efficient
> > Propagation and      |                          |   computation by caching
> > Matching             |                          |     intra-graph messages)
> > ---------------------------------------------------------------------------
> > Geometric Prior      |            No            |     Yes (by max margin
> >                      |                          |       lookahead loss)
> > ---------------------------------------------------------------------------
> > Asymmetric model     |            No            |         Yes (Eq. 4)
> > ---------------------------------------------------------------------------
> > Can be applied to    |   No (needs adaptation)  |            Yes
> > Subgraph Matching
> > ```

---

> > > ### Author Response · Authors · 2022-11-19
> > > **Additional Comparison**
> > >
> > > In addition, we list out some other differences between the two methods:
> > >
> > >
> > > ```
> > > Goal               |     Largest common    |       Entire of q must
> > >                    |       subgraph        |           be matched
> > > ---------------------------------------------------------------------------
> > > Allows some nodes  |            Yes        |            No
> > > in G_1 (or q)      |                       |
> > > to remain unmatched|                       |
> > > to G_2 (or G)      |                       |
> > > ---------------------------------------------------------------------------
> > > G1 (or q) size     | Can be extremely large|Usually orders-of-magnitude
> > >                    | and as large as G_2   |     smaller than G
> > > ---------------------------------------------------------------------------
> > > Search Framework   |        McSplit        |          Hybrid
> > > ---------------------------------------------------------------------------
> > > Preprocessing to   |           No          |     Yes (candidate set)
> > > reduce action space|                       |
> > > ---------------------------------------------------------------------------
> > > Backtrack Criterion|   Continue searching  |   End search branch if
> > >                    | when one node cannot  | any query node cannot
> > >                    |     be matched        |        be matched
> > > ---------------------------------------------------------------------------
> > > Condition of       |    Prune state if     |     Prune state if
> > > pruning states     |    largest possible   |   any node in the query
> > >                    |   matching is smaller |      cannot be matched
> > >                    |  than the current best|
> > >                    |         matching      |
> > > ---------------------------------------------------------------------------
> > > Isomorphism        | Matching nodes in the |  Matching nodes in the
> > > constraint         | same bidomain ensures | same local candidate set
> > > satisfaction       |       isomorphism     |   ensures isomorphism
> > >
> > > ```

---

> ### Author Response · Authors · 2022-11-15
> **Code uploaded as Supplementary Material**
>
> Dear Reviewer,
>
> We have uploaded our implementation as Supplementary Material.
>
> Thanks!

---

> ### Comment · Reviewer_vmC2 · 2022-12-12
> **Response to the authors rebuttal**
>
> Dear Authors,
>
> Thanks for the response.  I read the rebuttal and am still reluctant to increase the scores because, GMN also provide other model which is interaction free. There is no comparison with that. Moreover, I still think the delta between GLSearch and this paper is still not sufficient for ICLR.

---

> > ### Author Response · Authors · 2022-12-13
> > **Comparison to interaction-free version of GMN**
> >
> > **Summary**: We have compared against the interaction-free of GMN called “GIN-encoder” in Table 3 of Section D.1.1.
> >
> > **Part 1: GIN-encoder is equivalent to GMN’s interaction free version.**
> >
> > We assume the interaction-free version refers to the left figure of Figure 2 of https://arxiv.org/abs/1904.12787, which is defined in Section 3 as a standard GNN that only considers intra-graph and not inter-graph message passing. We compare QC-SGMNN to GIN, one of the most expressive standard GNN architectures, in Table 3 of Section D.1.1. In this experiment, all the components are identical to NSubS except for the encoder which is replaced with a 8-layer GIN. The decoder is also the same both settings returning node-node scores.
> >
> > Note, unlike GIN and GMN, we are among the first to (1) disentangle intra- and inter- graph messages and (2) perform inter-graph message passing based on search state.
> >
> > **Part 2: Experimentally QC-SGMNN outperforms GMN and interaction-free GNN encoder in Section D.1.1.**
> >
> > NSubS’s QC-SGMNN provides a large performance improvement compared to GMN, interaction-free GNN (GIN), as detailed in Table 3 of Section D.1.1.
> > ```
> >                   HPRD
> >                   Sol%  #Sol
> > NSubS             0.48  4.17
> > GMN-encoder       0.30  2.89
> > GIN-encoder       0.30  2.73
> > ```
> >
> > QC-SGMNN outperforms GIN, because it not only considers inter-graph message passing, but also leverages local candidate space in an efficient and effective manner. Again, we are among the first to propose a GNN that is a mixture of a static intra-graph message passing module that caches the embeddings (Section 3.2.1) and that does not change across search, and a dynamic inter-graph message passing module whose messages change across search and conditions the the model on the current search state.
> >
> > Furthermore, GMN and GIN do not solve the subgraph matching problem directly. They only provide a soft matching matrix, not hard subgraph matching. Without NSubS’ search framework and novel policy loss incorporating max-margin geometric priors, GMN and GIN cannot easily find hard subgraph matchings. Please see our “IsoNet experiments” section, which discusses why soft matching matrices are not enough  to solve subgraph matching (NSubS’ search framework is required). In the experiments above, GMN and GIN are in the NSubS’ framework, and hence more comparable to NSubS’ QC-SGMNN encoder than the whole RL model.

---

> > ### Author Response · Authors · 2022-12-13
> > **Comparison to GLSearch**
> >
> > **Summary:** GLSearch solves a different task, MCS detection, while this work solves Subgraph Matching. Due to the fact that Subgraph Matching requires matching of the entire query graph, and requires returning as many matchings as possible, the design of policy network for Subgraph Matching is more challenging, motivating the proposed novel QC-SGMNN encoder that captures the state of matching $s_t$ at each search iteration, which can double the performance of GLSearch.
> >
> > **Part 1: Subgraph Matching (SGM) is a practically more difficult task than Maximum Common Subgraph Detection (MCS).**
> >
> > Although SGM is a special case of MCS in theory, GLSearch is both trained and evaluated to find the largest common subgraph. Because MCS detection does not even have the notion of query graph or target graph, this MCS metric does not require the entirety of query graph to be matched against the target graph. In practice, as long as a large common subgraph is returned, a large partial reward is assigned. In contrast, SGM requires all nodes and edges in the query to be matched, and in practice, many solutions should be returned to the user. Therefore, how to design a successful framework under the “learning-to-search” paradigm for SGM is a more challenging task compared to MCS detection.
> >
> > **Part 2: GLSearch’s fundamentally produces static/state-independent node embeddings and cannot tackle the harder task of Subgraph Matching.**
> >
> > GLSearch fundamentally does *not* produce intra-graph message embeddings that are dynamic/adaptive to each state in the search process. GLSearch performs message passing for each input graph individually once before the search begins, i.e. **GLSearch lacks node embeddings that are adaptive to search states.** GLSearch resorts to a “factoring out action” trick to enlarge the effect of an action node pair. Specifically, it uses the fact that $Q(s_t, a_t) = 1 + V(s_{t+1})$, and performs a series of readout and interaction operations (decoder) on the static node embeddings to compute $Q(s_t, a_t)$. In contrast, NSubS assumes that richer node embeddings that dynamically reflect the state are beneficial for solving the harder task of SGM. In other words, we wish the candidate node embeddings to be state-dependent (dynamic) instead of state-independent (static) (Section 3.2.1).
> >
> > Another downside related to GLSearch’s lack of state-dependent node embeddings is that, the “factoring out action” scheme requires the sequential looping through candidate actions, $a_t$s, and computing the next states, $s_{t+1}$. Such costly sequential construction of next states is unavoidable, since the model must see the next states in order to predict the Q score for the action. The adapted version of GLSearch to the task of SGM inherits this downside, and we experimentally show the worse performance compared to NSubS.
> >
> > **Part 3: GLSearch’s node embeddings are not geometrically regularized to reflect the nature of SGM.**
> >
> > Another assumption of NSubS is that, geometrically regularized node embeddings trained with the max-margin geometric prior further benefit the task of SGM. NSM (https://arxiv.org/abs/2007.03092) and IsoNet (https://ojs.aaai.org/index.php/AAAI/article/view/20784) show incorporating geometric priors into node embeddings can drastically improve the performance of subgraph matching. NSubS’ lookahead loss unifies both geometric and RL loss to regularize its node embeddings. GLSearch does not consider any such regularization.
> >
> > **Part 4: GLSearch’s lack of state-dependent and geometrically regularized node embeddings cause its worse experimental results when adapted to the task of SGM.**
> >
> > Indeed, NSubS provides a large performance gain compared to GLSearch, as detailed in our “We provide experimental results for GMN, DMPNN, and GLSearch” response.
> > ```
> >               	HPRD
> >               	Sol%  #Sol
> > NSubS         	0.48  4.17
> > GLSearch-adapted  0.22  1.88
> > ```

---

### Author Response · Authors · 2022-11-09
**Common Question regarding novelty**

1. Encoder

* 1.1 Why is GMN not applicable for Subgraph Matching?

  * Graph Matching Neural Networks (GMN) performs node-to-node message passing between all the nodes in the two input graphs, resulting in quadratic time complexity, O(|V_q||V_G|). Such a naive adoption of GMN will not only be prohibitively inefficient for large graphs but also neglect the state-specific information, i.e. which nodes have been matched at each state s_t.

  * Moreover, GMN outputs a soft node-node matching matrix instead of a discrete matching result indicating which node in q should be matched to which node in G. To adopt GMN, we need to use GMN as the encoder and decode the node embeddings into search policy.

* 1.2 Our model: Query-Conditioned Subgraph Matching Neural Networks (QC-SGMNN)

  * Our major novelty is the incorporation of local candidate space A_{u’} into the neural network design, i.e. Equation (4) in the paper. The benefits of QC-SGMNN over GMN are two-fold. (1) It alleviates the computational burden by reducing from O(|V_q||V_G|) to O(|V_q||A_{u_t}|), where the average candidate space size is usually orders-of-magnitude smaller than the full target graph size |V_G|, e.g. |A_{u_t}| is around 10 for HPRD while |V_G|=9045. (2) QC-SGMNN produces state-dependent node embeddings via only performing message passing from A_{u_t}|.

* 1.3 GMN vs QC-SGMNN

  * 1.3.1 Besides the efficiency and efficacy improvements mentioned above, QC-SGMNN also performs graph-to-node message passing from q to nodes in G, whereas GMN only performs node-to-node message passing. We show the importance of this design in Section D denoted as “NSUBS-no-Q-matching”.

  * 1.3.2 Another innovation of QC-SGMNN is the decoupling of the inter-graph Matching module and the intra-graph Propagation module. Such decoupling allows for the caching of intra-graph embeddings computed once at the beginning of search and used throughout search to avoid recomputing the intra-graph message passing for each iteration.

* 1.4 Summary of novelty in encoder

  * 1.4.1 The key novelty is the integration of local candidate space which results in efficient incorporation of state-dependent information into node embeddings.

  * 1.4.2 The graph-to-node message passing in the inter-graph Matching module injects query graph information into the target graph node embeddings, which in turn interact with the query graph node embeddings through sequentially stacked layers.

  * 1.4.3 The decoupling of Matching and Propagation modules enables caching each Propagation module’s intermediate node embeddings, allowing for faster inference.

2. Loss

* 2.1 Why is NSM-style loss not enough?

  * The geometric loss proposed in https://arxiv.org/abs/2007.03092 and https://ojs.aaai.org/index.php/AAAI/article/view/20784 learn robust and interpretable representations for subgraph matching, but fails to consider the state-conditioned nature of search. Yet, subgraph matching on large graphs requires search to ensure constraint satisfaction.

* 2.2 Why is traditional RL loss not enough?

  * Traditional RL losses, such as REINFORCE, DQN, and PPO learn effective search aware state-dependent policies, but often overfit the training objective and get stuck in local minima and maxima.

  * To ensure it follows some geometric property, the representation for state and action has to be pretrained on existing geometric losses, which does not necessarily generalize well to our task, as pretraining does not account for the dynamic nature of search.

* 2.3 Summary of novelty in lookahead loss

  * 2.3.1 Our lookahead loss unifies both geometric and RL loss by formulating the geometric loss as a regularized policy gradient to solve the weaknesses of both approaches.

  * 2.3.2 Our lookahead loss improves the geometric loss proposed by prior work through making it state-conditioned under the Bellman Equation.

  * 2.3.3 Our method improves REINFORCE by regularizing it with proven geometric priors. Unlike REINFORCE, which only considers s_t and s_{t+1}, we look ahead in search, s_{t+k}, to obtain more training data for our geometric prior.

  * 2.3.4 The relationship between the REINFORCE, geometric, and lookahead loss can be summarized below:

    * 2.3.4.1 Geometric:	loss = policy_loss(s_0) + Regularization

      * (a) s_0 is always initially unmatched

      * (b) regularization is the max-margin loss

      * (c) (+) (-) pairs obtained from data sampling


    * 2.3.4.2 REINFORCE:	loss = policy_loss(s_t)

      * (a) s_t is dependent on currently matched subgraph

    * 2.3.4.3 Lookahead:	loss = policy_loss(s_t) + Regularization

      * (a) s_t is dependent on currently matched subgraph

      * (b) regularization is the max-margin loss

      * (c) (+) (-) pairs obtained from looking ahead in search

---

> ### Author Response · Authors · 2022-11-10
> **Common Question regarding novelty**
>
> 3. Decoder
>
> One novelty in the decoder design is the pooling of node embeddings from q instead of G. This is motivated by the fact that pooling leads to information loss, and that the QC-SGMNN encoder has performed interaction between q and G. Thus, the node embeddings of q have incorporated information from G, and since |V_q| << |V_G|, the pooling from q alleviates information loss and provides useful information for the decoder to leverage.

---

### Author Response · Authors · 2022-11-10
**Common Question regarding related work**

1. Graph Matching Network (GMN)

* 1.1 GMN introduces the idea of inter-graph message passing, but our novelties make it feasible, in terms of efficiency and effectiveness, for subgraph matching.

* 1.2 To improve GMN’s efficiency, QC-SGMNN (1) prunes the intergraph interaction space using the local candidate set and (2) decouples the intra- and inter- message passing for caching.*

* 1.3 To improve GMN’s effectiveness, QC-SGMNN (1) prunes noisy inter-graph interactions with the local candidate set, (2) proposes carefully designed graph-to-node propagation, and (3) is state-dependent (since it uses the local candidate set).*

2. Reinforcement Learning Based Query Vertex Ordering Model (RL-QVO)

* 2.1 RL-QVO is orthogonal to our work, proposing a better query vertex-ordering scheme for subgraph matching. For large graphs, target-vertex ordering becomes the bottleneck instead of query vertex-ordering, as enumerating all target vertices for each query vertex becomes infeasible by the NP-Hard nature of subgraph matching.

* 2.2 Unlike RL-QVO, which uses random target vertex-ordering, NSubS learns a policy network to perform target vertex-ordering.

* 2.3 In addition to RL-QVO’s single graph encoder, NSubS considers the query to target graph matching with QC-SGMNN.*

* 2.4 Unlike RL-QVO’s simple PPO loss, NSubS learns geometric node representations with lookahead loss.**

3. Neural Subgraph Match (NSM) and Interpretable Neural Subgraph Matching (IsoNet)

* 3.1 Both NSM and Isonet are trained to rank which query graph is most likely a subgraph of the target graph at a graph-level, not the discrete subgraph matching at a node-level.

* 3.2 NSM and IsoNet both utilize a geometric max-margin loss function, which we extend under the RL framework by (1) formulating the proposed loss function as a regularized policy gradient and (2) making the model state-dependent through QC-SGMNN.**

* 3.3 We adopt NSM node-based formulation over IsoNet’s edge-based formulation because the state-of-the-art search algorithm is node-based, and search is critical to ensuring the extracted subgraphs satisfy the isomorphism properties.

* 3.4 Both NSM and Isonet learn a soft matching matrix, not the discrete subgraph matching. While Isonet claims the Hungarian algorithm can be used to convert this soft matching matrix into a discrete subgraph matching. We experimentally verify this does not scale, please see "Experimental Results Regarding Isonet" response for details.

* 3.5 Model-wise, NSubS' QC-SGMNN uses state-dependent intergraph message passing, unlike NSM and IsoNet.*

4. Dual message passing (DMPNN)

* 4.1 DMPNN is orthogonal to our work, proposing an improved intra-graph message-passing algorithm. Their model is trained to count subgraph isomorphisms, whereas we directly find the subgraph matching. Their model requires expensive supervised data, whereas we don’t.

* 4.2 Unlike DMPNN’s single graph encoder, NSubS introduces an intergraph propagation module, QC-SGMNN, to handle graph pairs.*

5. Consistent Subgraph Matching (ConsGraph)

* 5.1 ConsGraph is orthogonal to our work, extending the subgraph matching task with more complicated node and edge constraints. The paper utilizes an existing subgraph matching algorithm, VF2 (https://ieeexplore.ieee.org/document/1323804), as a component to their larger system.

* 5.2 We argue NSubS is better than VF2, ConsGraph’s subgraph matching routine, as we beat more advanced versions of VF2 (https://dl.acm.org/doi/10.1145/3318464.3380581) in Figure 2 of our main paper. Hence, NSubS would improve the performance of ConsGraph if it replaces the VF2 subgraph matching routine with NSubS.

6. GLSearch

* 6.1 GLSearch is designed for maximum common subgraph rather than subgraph matching. Nevertheless, we improve many of the design decisions made by GLSearch: search framework, model design, and loss design.

* 6.2 Subgraph matching search frameworks are much more powerful than MCS search frameworks since they exploit the property that the whole subgraph must be matched.

* 6.3 NSubS needs to execute less actions per forward pass since it uses policy-based RL instead of value-based RL, hence the model and loss design is completely different. Policy-based methods inherently require less action executions.

* 6.4 NSubS’ node-node level GMN style intergraph message passing learns representations for multiple actions instead of GLSearch’s coarsening approach which only produces a value for executing one action.*

* 6.5 NSubS proposes a new policy network loss based on enforcing geometric priors in node representations. GLSearch’s loss does not have any geometric priors.**

*See the Encoder subsection of the Novelty common question for details.

**See the Loss subsection of the Novelty common question for details.

---

### Author Response · Authors · 2022-11-15
**We have uploaded our code.**

Dear Reviewers,

Thank you all for your comments and suggestions! We have uploaded our implementation as a zip file in the Supplementary Material section for reproducibility.

Thanks!

---

### Author Response · Authors · 2022-11-19
**Address of all related works**

Dear Reviewers,

We have updated our paper and addressed all of the related works. Please refer to Sections C and D for discussion and experimental results regarding the detailed comparison. We have also responded to you individually for the mentioned related works. Thank you very much for pointing out to these works to help us better clarify our proposed method.

For your convenience, we list all the related works and the pointers to the detailed comparison:

1. GMN: Section C.1 and Section D.1.2
2. DMPNN: Section C.2 and Section D.1.2
3. GLSearch: Section C.3 and Section D.1.2
4. IsoNet: Section C.4
5. RL-QVO: Section C.5
6. Consistent Subgraph Matching: Section C.5

Here are the baseline methods indexed by reviewers mentioning them:

1. By Reviewer vmC2: GMN, GLSearch, and IsoNet

* 1.1 GMN is referred to as https://arxiv.org/abs/1904.12787

* 1.2 GLSearch is referred to as https://arxiv.org/abs/2002.03129

* 1.3 IsoNet is referred to as https://ojs.aaai.org/index.php/AAAI/article/view/20784

2. By Reviewer 181W: Consistent Subgraph Matching, RL-QVO, IsoNet, DMPNN

* 2.1 Consistent Subgraph Matching is referred to as “Consistent Subgraph Matching over Large Graphs. ICDE 2022”

* 2.2 RL-QVO is referred to as “Reinforcement Learning Based Query Vertex Ordering Model for Subgraph Matching

* 2.3 IsoNet is referred to as “Interpretable Neural Subgraph Matching for Graph Retrieval. AAAI 2022”

* 2.4 DMPNN is referred to as “Graph Convolutional Networks with Dual Message Passing for Subgraph Isomorphism Counting and Matching. AAAI 2022”

Thank you again for your detailed reviews!

---

### Author Response · Authors · 2022-11-19
**We thank everyone for your feedback!**

We would like to express our deep gratitude for all your feedback which we have incorporated into our revisioned paper.

Authors of paper 4973

---

### Author Response · Authors · 2022-12-10
**Experimental Results Regarding Isonet**

IsoNet (https://ojs.aaai.org/index.php/AAAI/article/view/20784) learns to rank which query graph from a set of query graphs is most likely to be a subgraph in the target graph. Isonet is not designed to provide the subgraph matching, but does provide an interpretable soft matching matrix.

IsoNet argues the Hungarian algorithm can convert this soft matching matrix into a discrete subgraph matching, but this does not guarantee isomorphism. Hence, the algorithm may provide invalid solutions. We experimentally verify that valid subgraph matchings found by the Hungarian algorithm: (1) does not scale with query graph size and (2) does not scale with graph density.

*Isonet (original)*: In the original paper, the authors create a |Et| x |Et| matching matrix, between |Et| edges in the target and |Eq| edges in the query appended with |Et| - |Eq| “dummy” edges. They run the Sinkhorn algorithm on this matching matrix, followed by the Hungarian algorithm to obtain a discrete subgraph matching. We call this approach IsoNet (original). IsoNet (original) runs into memory errors when the target graph has more than 10,000 edges. Meanwhile, NSubS scales to target graphs with over 2,900,000 edges.

*Isonet (scalable)*: To test the Hungarian algorithm on larger datasets, we run IsoNet (original) on a |Et| x |Eq| matching matrix, between |Et| edges in the target and |Eq| edges in the query, instead of the |Et| by |Et| matching matrix. Because this makes IsoNet scale linearly instead of quadratically with |Et|, we call this method IsoNet (scalable).

Both Isonet (original) and Isonet (scalable) use IsoNet’s official codebase: https://github.com/Indradyumna/ISONET.

We experimentally verify that, unlike NSubS, both IsoNet (original) and IsoNet (scalable) do not scale well with graph density and query graph size. NSubS scales better because its search process guarantees extracted subgraph matchings are isomorphic. In the below experiments, we generate datasets of Barabasi-Albert graphs under different settings. For each setting, 50 graph pairs are generated. We follow the same query graph sampling scheme as reported in Section A.2 of our main paper. We consider a pair solved if the algorithm finds an isomorphic subgraph mapping. We report the % of the 50 graph pairs solved for each dataset.

We first show both IsoNet (original) and IsoNet (scalable) do not scale with graph density. Here, we use a target graph with 32 nodes; and query graphs with 4, 8, and 16 nodes; and attachment parameter* of 2. Notably, IsoNet (scalable) performs better than IsoNet (original), since the dummy nodes introduce extra noise to the matching process.
```
                   BA (|Vq|=4)  BA (|Vq|=8)  BA (|Vq|=16)
IsoNet (original)  46%          8%           0%
IsoNet (scalable)  80%          4%           0%
```

Next, we show both IsoNet (original) and IsoNet (scalable) do not scale with query graph size. Here, we use a target graph with 32 nodes; and query graphs with 8 nodes; and attachment parameter* of 1, 2, 3.
```
                   BA (a*=1)  BA (a*=2)  BA (a*=3)
IsoNet (original)  60%        8%         0%
IsoNet (scalable)  82%        4%         0%
```

We show NSubS outperforms both IsoNet (original) and IsoNet (scalable). Following Table 4 in our main paper, we use a target graph with 10000 nodes; and query graphs with 8, 16, 32, and 64 nodes; and attachment parameter* of 3. OoM denotes “out of memory”. NSubS is run for 5 minutes on each graph pair. Figure 2.a in our main paper show results under different runtime cutoffs. Table 4 of our main paper show the performance of other search-based baselines.
```
                   BA-8   BA-16  BA-32  BA-64
IsoNet (original)  OoM    OoM    OoM    OoM
IsoNet (scalable)  40%    26%    4%     0%
NSubS              100%   100%   98%    62%
```

NSubS finds solutions to more graph pairs than IsoNet. Furthermore, since the Hungarian algorithm returns only a single subgraph matching, IsoNet can only produce one subgraph matching. Thus, NSubS also finds more subgraph matchings for each solved pair than IsoNet, as seen in Table 4 of our main paper. Again, NSubS outperforms IsoNet since NSubS learns a model in the context of search, which is guaranteed to output subgraph matchings that are isomorphic.

For more differences between NSubS and IsoNet, please refer to part 3 of the "Common Question regarding related work" response.

---

### Author Response · Authors · 2022-12-12
**Summary of reviews and our response**

We summarize the main concerns raised by reviewers:

1. Novelty

Concern: The novelty is limited.

Our response: We propose a Query-Conditioned Subgraph Matching Neural Networks that performs state-dependent inter-graph message passing for subgraph matching, and propose a novel lookahead loss that is both state-dependent and regularized with geometric priors.

2. Runtime overhead

Concern: The initial overhead is large and the proposed method only outperforms baselines after a few minutes.

Our response: Figure 7 in Section D.3 shows the proposed method finds the solution in very few search iterations, demonstrating the effectiveness of the learned policy over the baselines. Even with the initial overhead, our current results are still better considering performance vs clock time (Figure 2). In future, the initial overhead in running time can be countered by efficient implementation or techniques such as knowledge distillation for GNNs (Zhang, Shichang, et al. "Graph-less neural networks: Teaching old mlps new tricks via distillation." arXiv preprint arXiv:2110.08727 (2021).).

3. Comparison to existing works

Concern: Comparison to the following existing works are insufficient or missing: GMN, DMPNN, GLSearch, IsoNet, RL-QVO, and Consistent Subgraph Matching

Our response: We have addressed all of the mentioned related works:

Category 1: Works tackling different tasks

GMN: Section C.1 and Section D.1.2. GMN is designed for global graph-graph similarity score computation, and performs message passing between all the nodes in the two input graphs which are too expensive and do not consider each state’s specific matching status.

GLSearch: Section C.3 and Section D.1.2. GLSearch is designed for Maximum Common Subgraph detection. It proposes a value network to substitute the policy network for an action node pair, and we experimentally show the adaptation of GLSearch to Subgraph Matching leads to poorer performance, suggesting the need to design a model specifically for the task of Subgraph Matching.

Category 2: Works tackling Subgraph Matching

DMPNN: Section C.2 and Section D.1.2. DMPNN introduces a new message-passing mechanism but does not consider the information contained in a state of matching in the search process.

IsoNet: Section C.4. IsoNet matches the two graphs in one shot without search, and thus cannot guarantee the resulting subgraphs to ensure the isomorphism constraint of the task.

RL-QVO: Section C.5. RL-QVO proposes learning to select nodes from the query graph instead of the target graph, which is an orthogonal direction. RL-QVO leverages the fact that every node in the query graph must be matched, and thus learns a global node ordering between all the query graph nodes, and thus cannot be applied to our case of selecting target graph nodes that are computed dynamically at each state.

Consistent Subgraph Matching: Section C.5. This work essentially extends any existing subgraph matching algorithm to handle more complicated node and edge constraints, e.g. the matched subgraph must satisfy the constraint that it includes a “teacher” node and that teacher node must have two neighbors with the label of “course”.


Thank you for your consideration.

Sincerely,

Authors

---

### Decision · Program_Chairs · 2023-01-20

**Decision:**

Reject

**Justification For Why Not Higher Score:**

The reasons are stated in the meta-review. Lack of related works and strong baselines, incremental to an existing work. Though the authors did hard work during the rebuttal, their responses did not convince the reviewers in the end.

**Justification For Why Not Lower Score:**

N/A

**Metareview: Summary, Strengths And Weaknesses:**

The paper studies the subgraph matching problem where one needs to determine whether a query graph appears in a given target graph. The paper proposes a graph neural network-based encoder-decoder architecture combined with a reinforcement learning strategy. The GNN consists of two main modules: intra-graph message passing for modeling graph structures, and inter-graph message passing for query-graph matching. A look-ahead loss is used to supplement training signals.

Through a reviewer-AC meeting, reviewers point out several main concerns for the paper. First, there are many related works missing in their submitted version, which is pointed out by 3/4 reviewers in their reviews. Secondly, some powerful baselines are missing such as GMN. Reviewers point out that GMN is actually very hard to beat and there is no excuse to not compare. Thirdly, the paper is incremental to an existing work GLSearch. While I appreciate the hard work the authors did during the rebuttal, at the current stage I decided to respect the reviewers' opinions and recommend a rejection.

**Summary Of Ac-Reviewer Meeting:**

In the AC-reviewer meeting, reviewers and I reached a consensus that this paper still does not meet the acceptance bar. In particular, it misses a lot of related works and does not compare with some strong baselines. It is also incremental to an ICML 2022 paper studying a similar topic. Given one of the reviewers work particularly on this subfield while I do not, I decided to trust their opinion.